# CAST: Cluster-Aware Self-Training for Tabular Data

## Abstract

Self-training has gained attraction because of its simplicity and versatility, yet it is vulnerable to noisy pseudo-labels. Several studies have proposed successful approaches to tackle this issue, but they have diminished the advantages of self-training because they require specific modifications in self-training algorithms or model architectures. Furthermore, most of them are incompatible with gradient boosting decision trees, which dominate the tabular domain. To address this, we revisit the cluster assumption, which states that data samples that are close to each other tend to belong to the same class. Inspired by the assumption, we propose **C**luster-**A**ware **S**elf-**T**raining (**CAST**) for tabular data. CAST is a simple and universally adaptable approach for enhancing existing self-training algorithms without significant modifications. Concretely, our method regularizes the confidence of the classifier, which represents the value of the pseudo-label, forcing the pseudo-labels in low-density regions to have lower confidence by leveraging prior knowledge for each class within the training data. Extensive empirical evaluations on up to 21 real-world datasets confirm not only the superior performance of CAST but also its robustness in various setups in self-training contexts.

## 1 Introduction

Self-training is a simple and versatile semi-supervised learning method as it is easily adaptable for universal model architectures or training algorithms. It is an iterative algorithm that trains a classifier using a pseudo-labeling procedure, which assigns pseudo-labels to unlabeled data to use as labeled data to minimize entropy in each iteration. Contemporary self-training methods consider the confidence, often referred to as prediction probabilities of the classifier, as the score and generate a pseudo-label if the confidence score is higher than or equal to a certain threshold (Xie et al., 2020b; Pham et al., 2021). Therefore, the confidence, which represents the value of the pseudo-label, is a key component of self-training. However, it may not consistently serve as a reliable metric in real-world scenarios for various reasons such as biased classifiers or overconfidence in neural networks (Guo et al., 2017). These erroneous confidence scores can lead to the generation of noisy pseudo-labels during the self-training iterations, which may introduce confirmation bias that undermines the final self-training performance (Arazo et al., 2020). Given these potential pitfalls, relying solely on the confidence may be a precarious choice (Zou et al., 2019; Rizve et al., 2021; Xu et al., 2023).

Various studies have proposed solutions to counteract the noise in pseudo-labels induced by erroneous confidence, but they have diminished the simplicity and versatility of self-training. Concretely, they often necessitate modifications to self-training algorithms or alterations in the model architectures (Li & Zhou, 2005; Tanha et al., 2017; Rizve et al., 2021; Seibold et al., 2022). Furthermore, most of them are not applicable to gradient boosting decision trees (GBDT) as they are designed for neural networks. These limitations pose a substantial impediment to practitioners who want to apply reliable self-training on the tabular data where GBDTs have been the dominant architectures (Kaggle, 2021; Borisov et al., 2022; Shwartz-Ziv & Armon, 2022). Therefore, we conclude that any enhanced self-training for the tabular domain must maintain simplicity and versatility. Consequently, we study a natural but ignored question: *Can we improve self-training for tabular data by making confidence more reliable, without altering the self-training algorithm or model architecture?*

Several studies have been conducted to improve confidence more reliable without modifying existing algorithms. Specifically, they aim to make the confidence of the classifier reflecting its ground

truth correctness likelihood for safe decisions by calibrating the confidence using post-processing techniques (Guo et al., 2017; Wenger et al., 2020; Gupta et al., 2020). However, when applied to self-training in the tabular domain, an intriguing question arises: *Does well-calibrated confidence denote reliable confidence in the self-training context?*

Contemporary pseudo-labeling techniques for self-training approaches are divided into two primary strategies: fixed-threshold pseudo-labeling and curriculum pseudo-labeling. Within fixed-threshold pseudo-labeling strategies, pseudo-labels are designated once their confidences meet or exceed a certain threshold (Tur et al., 2005; Zoph et al., 2020; Xie et al., 2020a). Meanwhile, curriculum pseudo-labeling strategies generate pseudo-labels based on a threshold but operate under the premise that samples with higher confidence are easier for the classifier to handle. The classifier initially focuses on these "easier" pseudo-labels and, over time, progressively addresses more complex samples by incrementally lowering the threshold (Cascante-Bonilla et al., 2021; Zhang et al., 2021a). Considering the tabular domain, where the predominant architecture, GBDTs, necessitates hard pseudo-labels, the extent to which the confidence exceeds the threshold is meaningless for both strategies. Hence, given the above premise of curriculum pseudo-labeling and consideration, we conclude the key components of reliable confidence in the self-training context as follows: *(1) lowering the confidences of unreliable pseudo-labels below a threshold* and *(2) reflecting how easy it is for the classifier*.

After dissecting self-training, we argue that the *cluster assumption*, foundational to semi-supervised learning (SSL), can guide to trustworthy confidence in self-training. The cluster assumption states that the data points nearby are likely to belong to the same class. As such, the decision boundary should avoid high-density regions, favoring low-density regions instead (Chapelle & Zien, 2005; Wang et al., 2012; Lee et al., 2013). Therefore, by assigning high confidence to pseudo-labels in high-density regions and low confidence to those in low-density regions, the confidences ensure that reliable pseudo-labels remain above the threshold and reflect how easy pseudo-labels are for the classifier.

In this study, we propose **CAST**: **C**luster-**A**ware **S**elf-**T**raining for tabular data. CAST regularizes the confidence during the pseudo-labeling procedure by reflecting the cluster assumption utilizing the local density of the unlabeled sample. Consequently, CAST leads to performance gain without significant modifications to existing self-training algorithms or model architectures. Note that CAST aims to lower the confidence of the pseudo-labels in low-density regions, while confidence calibration methods aim to mirror the true likelihood.

Our key contributions are summarized as follows: (1) We propose CAST, a novel cluster-aware self-training approach for tabular data. To the best of our knowledge, this is the first attempt of enhancing self-training solely by refining confidence more reliable in the self-training context. (2) Unlike previous reliable pseudo-labeling techniques that require special requirements, our method seamlessly integrates with current self-training algorithms and tabular models. (3) Our extensive experiments on up to 21 real-world classification datasets confirm that regularized confidence of CAST consistently delivers marked performance enhancements across various setups, while calibrated confidence is meaningless in self-training contexts.

## 2 RELATED WORKS

**Reliable Pseudo-Labeling for Self-Training.** Reliable pseudo-labeling has attracted considerable interest in self-training contexts. One of the primary approaches to reliability is noise filtering. For example, Li & Zhou (2005) and Wang et al. (2010) use cut edge weights to eliminate noisy pseudo-labels to ensure reliable pseudo-labeling. Zhou et al. (2012) create subsets of unlabeled data using the distance to the decision boundary of each subset to discern and retain useful subsets while discarding those deemed unreliable. Gan et al. (2013) employ clustering analysis to eliminate unreliable samples. In addition to noise filtering, there are other studies for reliable pseudo-labeling. Tanha et al. (2017) demonstrate not only distance-based noise filtering, but also enhancements to decision trees for self-training. Zou et al. (2019) regularize the confidences and use them as soft pseudo-labels to prevent infinite entropy minimization. Zhang et al. (2021b) suggest online denoising of pseudo-labels based on their approach to the relative feature distances to a prototype, which means the feature centroids of classes. Rizve et al. (2021) present an uncertainty-aware pseudo-label selection framework that improves pseudo-labeling accuracy. Yang et al. (2022) propose a self-training

framework that performs selective re-training by prioritizing reliable pseudo-labels based on holistic prediction-level stability. Chen et al. (2022) introduce a debiased self-training that avoids the accumulation of errors during self-training iteration owing to the bias. Seibold et al. (2022) use a small number of labeled data as reference and selected pseudo-labels that have the semantics of the best fitting in a reference set. Niu et al. (2022) ensure the reliability of pseudo-labels through the use of a semantically consistent ratio, while Li et al. (2022) enhance clustering performance by selectively incorporating the most confident predictions from each cluster. Recently, Xu et al. (2023) adopt a neighborhood-based sample selection approach, which is guided by data representation to refine pseudo-labels. However, most of their work requires significant modifications to conventional self-training algorithms or model architectures, with several showing incompatibilities with GBDTs.

**Confidence Calibration.** Poorly calibrated confidence is one of the most prevalent problems in various models (Caruana et al., 2004; Guo et al., 2017; Wang et al., 2021). Guo et al. (2017) define that a classifier is well-calibrated when its confidence estimates are representative of the true correctness likelihood. This definition has been widely accepted across various studies (Mukhoti et al., 2020; Gupta et al., 2020; Wenger et al., 2020; Hebbalaguppe et al., 2022; Liu et al., 2022). One of the most widely used metrics for calibration to measure how well the classifier is calibrated is Expected Calibration Error (ECE) [1] (Naeini et al., 2015). There are two primary strategies for achieving a well-calibrated model that produces reliable confidence. The first approach aims to calibrate the classifier during training (Mukhoti et al., 2020; Hebbalaguppe et al., 2022; Liu et al., 2022), whereas the second performs post-hoc calibration by transforming the confidence of a given classifier (Gupta et al., 2020; Wenger et al., 2020). However, it is noteworthy that achieving a well-calibrated classifier is not without potential trade-offs; some studies suggest that while enhancing calibration, accuracy might be inadvertently compromised (Wang et al., 2021; Zhu et al., 2022). Moreover, the inherent value of the calibration applied in self-training remains underexplored[2], although certain calibration techniques incidentally improve both the calibration and performance of self-trained classifiers (Wang et al., 2021; Munir et al., 2022).

# 3 CAST: CLUSTER-AWARE SELF-TRAINING

To improve self-training through reliable confidence, we revisit the cluster assumption, which is a fundamental assumption in semi-supervised learning. The assumption posits that data samples that are close to each other tend to belong to the same class, and that decision boundaries should lie in low-density regions (Chapelle & Zien, 2005; Wang et al., 2012; Van Engelen & Hoos, 2020). This concept implies that the pseudo-labels that lie in high-density regions are more reliable than those that lie in low-density regions. The empirical results shown in Figure 1 also support that the cluster assumption should be considered in pseudo-labeling. Inspired by the assumption, we conclude that pseudo-labels in low-density regions should have lower confidence than those in high-density regions. Therefore, we propose CAST for tabular data to lower the confidence of pseudo-labels lying in low-density regions. Concretely, CAST regularizes the confidence during pseudo-labeling

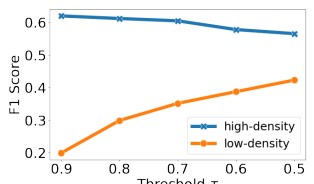

Figure 1: F1 score of pseudo-labels across high- and low-density regions over confidence threshold $\tau$ on 6M mortality dataset[3]

procedure using prior knowledge for each class from the training data. We show the regularized pseudo-labeling procedure of CAST in Section 3.1 and the full algorithm of CAST in Section 3.2.

## 3.1 REGULARIZED PSEUDO LABELING

Given $i^{th}$ unlabeled data $\mathbf{x}^{(i)}$, pseudo-label $\tilde{\mathbf{y}}^{(i)} = [\tilde{y}_1, \tilde{y}_2, ..., \tilde{y}_{N-1}, \tilde{y}_N]$ for $N$-class dataset is generated based on the confidence $\mathbf{c} = [c_1, c_2, ..., c_{N-1}, c_N]$, which the classifier produces for given $\mathbf{x}^{(i)}$, where

$$\tilde{y}_j = \left\{ \begin{array}{ll} 1 & \text{if } j == \text{argmax}(\mathbf{c}) \text{ and } \max(\mathbf{c}) >= \tau \\ 0 & \text{otherwise} \end{array} \right\} \tag{1}$$

---

[1] Expected Calibration Error, refer to Appendix A for more details.

[2] For a comprehensive discussion on this topic, refer to Appendix B.

[3] We generate pseudo-labels using XGBoost (Chen & Guestrin, 2016). Then, we estimate the density using empirical likelihood and split the top 50% as high-density, and the rest as low-density.

In eq (1), a pseudo-label is generated to be the class with the highest confidence if the confidence surpasses the specific threshold, $\tau$. As pseudo-labels in low-density regions are unreliable, we have to reduce the confidence of pseudo-labels that lie in low-density regions. We get the estimated density for unlabeled samples by extracting the prior knowledge using a density estimator $D_t$ (e.g. multivariate kernel density estimator or empirical likelihood) which is fitted to the labeled training data distribution $t$ [4]. Here, the prior knowledge $\gamma$ for each class is defined as follows:

$$\gamma^{(i)} \leftarrow D_t(\mathbf{x}^{(i)}), \quad \text{where} \quad \gamma^{(i)} = [\gamma_1, \gamma_2, ..., \gamma_{N-1}, \gamma_N] \qquad (2)$$

Then, we normalize $\gamma$ using a min-max scaler because the scale of $\gamma$ varies among implementations, and we need a relative measure to align unlabeled samples. To make pseudo-labels in low-density regions have lower confidence, we have to adjust the magnitude of $\mathbf{c}$ according to the prior knowledge. Element-wise product $\gamma$ to $\mathbf{c}$ can achieve this, such as follows:

$$\gamma \circ \mathbf{c} \qquad (3)$$

However, prior knowledge is usually incomplete, particularly in semi-supervised learning settings where the labeled training data is scarce. To regulate the influence of prior knowledge on pseudo-label valuation, we adjust the balance between eq (3) and $\mathbf{c}$ using the hyperparameter $\alpha$. The regularized pseudo-labeling procedure of CAST is defined as follows:

$$\tilde{y}_j = \left\{ \begin{array}{ll} 1 & \text{if } j = \operatorname{argmax}(f(\mathbf{c})) \text{ and } \max(f(\mathbf{c})) >= \tau \\ 0 & \text{otherwise} \end{array} \right\}, \text{ where } f(\mathbf{c}) = \alpha(\gamma \circ \mathbf{c}) + (1-\alpha)\mathbf{c} \quad (4)$$

In this formulation, $f$ is the scoring function of CAST, which evaluates pseudo-label not only considering the confidence of the classifier but also prior knowledge. The hyperparameter $\alpha$ delineates the influence of prior knowledge on pseudo-label valuation. If $\alpha$ is close to 0, it leads to the pseudo-labeling procedure that uses only the confidence to decide whether to generate the pseudo-label for given $\mathbf{x}$, which is the same pseudo-labeling procedure as the one used in the conventional self-training. Conversely, a high $\alpha$ value, approaching 1, steers the pseudo-labeling procedure to prioritize $\gamma \circ \mathbf{c}$.

**Discussion.** Note that the only difference between CAST and the conventional self-training algorithm is whether the use of regularized confidence (eq (4)) or naive confidence (eq (1)) to evaluate the pseudo-labels. Therefore, CAST retains the simplicity and versatility of self-training and is also compatible with conventional self-training algorithms, and various models in the tabular domain.

## 3.2 ALGORITHM OF CAST

---

**Algorithm 1** CAST

---

**Input:** Labeled and unlabeled dataset $D_L$ and $D_U$; pseudo-labeling algorithm $\Phi$ which adopt eq (4); target classifier $C$; performance metric $P$.
**Output:** The best classifier during the self-training iterations, $C_{best}$.

$C_{current} \leftarrow$ trained classifier on $D_L$
$C_{best} \leftarrow C_{current}$
**while** the termination conditions of $\Phi$ are not met **do**
  $\tilde{D} \leftarrow D_L$
  **for** $\mathbf{x}^{(i)}$ in $D_U$ **do**
    $\mathbf{c} \leftarrow C_{current}(\mathbf{x}^{(i)})$
    $\tilde{\mathbf{y}}^{(i)} \leftarrow \Phi(\mathbf{c})$
    **if** $\tilde{\mathbf{y}}^{(i)} \neq \vec{0}$ **then**
      $\tilde{D} \leftarrow \tilde{D} \cup \{(\mathbf{x}^{(i)}, \tilde{\mathbf{y}}^{(i)})\}$
  $C_{current} \leftarrow$ a classifier newly trained on $\tilde{D}$
  **if** $P(C_{current}) > P(C_{best})$ **then**
    $C_{best} \leftarrow C_{current}$
**Return:** $C_{best}$

---

Let $D_L = \{(\mathbf{x}^{(i)}, \mathbf{y}^{(i)})\}_{i=1}^{N_L}$ denote a labeled dataset consisting of $N_L$ samples for an $N$-class classification task. Here, $\mathbf{x}^{(i)}$ represents the features of the $i^{th}$ sample and $\mathbf{y}^{(i)}$ is its corresponding label. Similarly, let $D_U = \{(\mathbf{x}^{(i)}, \varnothing)\}_{i=1}^{N_U}$ denote an unlabeled dataset comprising $N_U$ samples, each characterized solely by its features $\mathbf{x}^{(i)}$. Furthermore, we represent a subset of $D_U$ as $\tilde{D}_U$, and the size of $\tilde{D}_U$ as $\tilde{N}_U$. For every unlabeled sample, a pseudo-label $\tilde{\mathbf{y}}^{(i)}$ is produced by the pseudo-labeling algorithm $\Phi$ after the classifier $C$ generates the confidence, $\mathbf{c}$, for given $\mathbf{x}^{(i)}$. Here, $\Phi$ represents the pseudo-labeling algorithm (such as fixed-threshold or curriculum pseudo-labeling) that employs eq (4) instead of eq (1) typically used in conventional

self-training algorithms. Finally, $\tilde{D} = \{(\mathbf{x}^{(i)}, \mathbf{y}^{(i)} \text{ or } \tilde{\mathbf{y}}^{(i)})\}_{i=1}^{N_L + \tilde{N}_U}$ signifies the combined training set used for every self-training iteration, encompassing both $D_L$ and the pseudo-labeled subset $\tilde{D}_U$ containing $\tilde{N}_U$ samples. Algorithm 1 shows the full algorithm of CAST.

---

[4]The natural characteristic of the tabular data is each feature occupies a specific, fixed position within the table. This allows us to directly extract prior knowledge from the labeled training dataset unlike other domains (e.g. image or text). The specific choice of density estimator for CAST depends on the implementation.

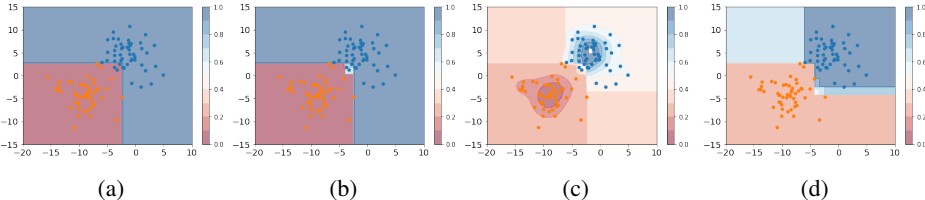

Figure 2: Visualization of the confidence levels of XGBoost on the Blob dataset when generating pseudo-labels for the third self-training iteration with FPL using (a) naive confidence, (b) calibrated confidence with HB, (c) regularized confidence with CAST-D, and (d) regularized confidence with CAST-L. Colored points represent labeled samples in the training set for each class, and the degree of the color indicates the confidence level in the space where the Blob data exists.

## 4 EXPERIMENTAL EVALUATION

In this section, we design a suite of experiments to answer the questions that we raised in Section 1 as follows: (1) *Can we improve self-training for tabular data by making confidence more reliable, without altering the self-training algorithm or model architecture?* (2) *Does well-calibrated confidence denote reliable confidence in the self-training context?*

The experimental procedure consists of three distinct steps:

1. We visualize and analyze the impact of diverse confidence on self-training using a toy dataset. This is further elaborated in Section 4.1.

2. We present empirical results in the context of self-training with diverse confidence using real-world tabular datasets in Section 4.2.

3. We conclude our experiments with additional analyses of CAST, scrutinizing several aspects of CAST, as discussed in Section 4.3.

For all the experiments, we establish a baseline using naive confidence-based self-training. Within our notation, fixed-threshold pseudo-labeling is denoted as FPL, and curriculum pseudo-labeling is referred to as CPL. Unless otherwise noted, we use the following settings. We empirically adopt a threshold, $\tau$, of 0.6 for FPL. For CPL, we set the starting threshold to capture the top 20% and incrementally increase the percentage by 20%, in line with the recommendations of Cascante-Bonilla et al. (2021). Self-training iterations are terminated under two conditions: for FPL, when a self-trained classifier underperforms after self-training iteration, and for CPL when no additional unlabeled data remain. To mitigate confirmation bias accumulation during self-training iterations, we reinitialize all classifiers after generating pseudo-labels, as recommended by Cascante-Bonilla et al. (2021).

Given the prevalence of GBDTs in the tabular domain, we focus on model-agnostic post-hoc calibration methods. We choose temperature scaling and histogram binning for the confidence calibration because of their simplicity and widespread use (Guo et al., 2017). We also use spline (Gupta et al., 2020) and latent Gaussian process (Wenger et al., 2020) calibrations for more sophisticated calibrations. We adopt a multivariate kernel density estimator and empirical likelihood as a density estimator to derive prior knowledge. The implementation details of prior knowledge are in Appendix D. For clarity, we use the following abbreviations: temperature scaling (TS), histogram binning (HB), spline calibration (SP), and latent Gaussian process (GP). Our proposed CAST methods, with a multivariate kernel density estimator and empirical likelihood are denoted as CAST-D and CAST-L, respectively.

### 4.1 TOY DATASET

#### 4.1.1 DATASET AND IMPLEMENTATION DETAILS.

To demonstrate the effects of various confidences in self-training, we create a binary classification toy dataset, *Blob*, using the `scikit-learn` package (Pedregosa et al., 2011). This dataset consists

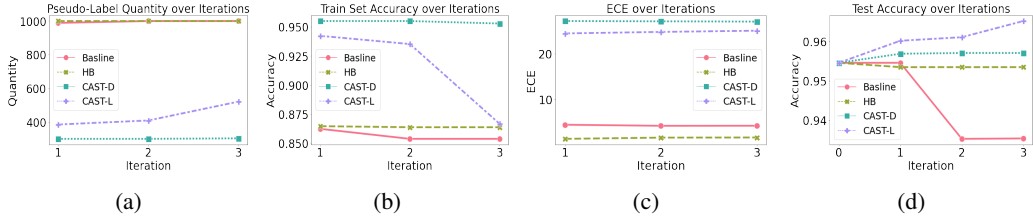

Figure 3: Pseudo-label quantity, training set accuracy, test ECE, and test accuracy, in sequence, for each confidence-based self-training over iterations.

of 100 training, 1,000 validation, 10,000 test, and 1,000 unlabeled samples designated for self-training. We employ the XGBoost classifier (Chen & Guestrin, 2016), and the hyperparameters are optimized using Optuna (Akiba et al., 2019) over 50 trials. Subsequently, we conduct four distinct self-training approaches, each with three iterations of FPL. Each approach employs naive confidence, calibrated confidence with HB, regularized confidence with CAST-D, and regularized confidence with CAST-L.

### 4.1.2 Results and Analysis.

Figure 2 presents an overlay of the training data and confidence levels for each classifier. The confidences of CAST exhibit reduced confidences for samples that lie in low-density regions as illustrated in Figure 2 (c) and (d). Contrarily, the naive confidence and calibrated confidence of HB do not differentiate confidence levels between high and low-density regions (Figure 2 (a) and (b)). Figure 3 shows a comparison of the pseudo-label quantity, training set accuracy, and ECE when generating pseudo-labels for each self-training iteration along with the test accuracy across every self-training iteration. In this figure, it is observed that the baseline is prone to confirmation bias, leading to diminished performance after three self-training iterations. Although HB records the lowest ECE over the iterations, a mere reduction in ECE does not guarantee accurate pseudo-labels or enhanced performance in self-training. However, our CASTs exhibit improved performance with reliable pseudo-labels by lowering the confidence of unreliable pseudo-labels, although they display a notably higher ECE.

## 4.2 Empirical Evaluation

### 4.2.1 Datasets and implementation details.

To empirically evaluate the different confidences in self-training, we use four tabular datasets with XGBoost (Chen & Guestrin, 2016), FT-Transformer (Gorishniy et al., 2021), and MLP. First, we adopt the 6-month mortality prediction post-acute myocardial infarction (in short, 6M mortality) dataset from the Korea Acute Myocardial Infarction Registry (KAMIR). The scarcity of labels in the dataset inspired us to study self-training in the tabular domain. The other three datasets (diabetes, ozone, and cmc) are sourced from OpenML-CC18—a benchmark suite of meticulously curated datasets (Vanschoren et al., 2014; Bischl et al., 2017; Feurer et al., 2021). Our choice of these datasets aims to illustrate the impact of CAST across diverse data domains. We also conduct extended empirical experiments using an additional seventeen datasets from OpenML-CC18 with XGBoost to demonstrate the results for broader datasets, which are reported in Appendix I.

We evaluate the performance based on the relative improvement compared with a supervised classifier. This approach is adopted because appropriate metrics can vary across datasets, and the primary objective of SSL is to measure its advantages over supervised settings (Oliver et al., 2018). Relative improvement is assessed using the F1-score for both the 6M mortality and ozone datasets, accuracy for the diabetes dataset, and balanced accuracy for the cmc dataset. Given that the ultimate goal of SSL is to surpass the performance of well-tuned supervised models (Oliver et al., 2018), we optimize each model using Optuna (Akiba et al., 2019) for over 100 trials. This optimized model serves dual purposes: it provides a baseline performance to gauge the relative improvements achieved through self-training and is used as a base classifier for self-training. As noted by (Oliver et al., 2018; Su et al., 2021), relying solely on an insufficient validation set can lead to suboptimal hyperparameter

Table 1: Relative improvement over four tabular datasets. The top results are highlighted in bold, while the second-best scores are underlined. Abbreviations are as follows: temperature scaling (TS), histogram binning (HB), spline calibration (SP), and latent Gaussian process (GP).

|  |  | 6M mortality | | | diabetes | | | ozone | | | cmc | | |
|---|---|---|---|---|---|---|---|---|---|---|---|---|---|
|  |  | XGB | FT | MLP | XGB | FT | MLP | XGB | FT | MLP | XGB | FT | MLP |
| FPL | Baseline | 4.090 | 1.123 | 7.878 | 0.000 | 0.333 | 1.301 | 0.354 | 0.336 | 1.284 | 0.774 | 0.251 | 0.143 |
|  | TS | 4.090 | 1.123 | 7.699 | 0.000 | 0.333 | 1.301 | 0.354 | 0.336 | 1.284 | 0.774 | 0.251 | 0.143 |
|  | HB | 4.126 | 0.000 | -0.142 | 0.032 | 1.000 | 0.787 | -0.566 | 2.523 | -0.149 | 0.311 | 1.032 | 0.996 |
|  | SP | 4.266 | 2.315 | 8.444 | -0.098 | 0.212 | 0.8778 | -0.384 | 5.017 | 2.574 | -0.214 | 1.411 | 0.000 |
|  | GP | 1.087 | -1.117 | -0.069 | 0.786 | 0.788 | -0.212 | 1.091 | 1.004 | -1.426 | 0.000 | 0.000 | 0.000 |
|  | CAST-D | 4.091 | 5.562 | 10.542 | 1.604 | 1.394 | 1.725 | 7.331 | 8.860 | 9.055 | 2.325 | 0.716 | 1.612 |
|  | CAST-L | 9.597 | 8.951 | 16.981 | 1.342 | 0.667 | 1.967 | 6.588 | 6.729 | 8.056 | 2.363 | 1.783 | 1.046 |
| CPL | Baseline | -0.652 | 6.105 | 4.852 | 0.131 | 0.818 | 0.787 | 4.986 | -2.276 | 5.878 | 0.850 | 1.922 | 0.423 |
|  | TS | -0.652 | 6.105 | 4.852 | 0.131 | 0.818 | 0.787 | 4.986 | -2.276 | 5.640 | 0.850 | 1.922 | 0.423 |
|  | HB | 4.843 | 2.269 | 6.820 | 0.131 | 1.636 | 0.363 | 0.579 | 2.017 | 0.454 | 0.718 | 1.171 | 0.219 |
|  | SP | -0.777 | 6.306 | 8.320 | -0.164 | 1.424 | 0.393 | 1.998 | 1.286 | 5.713 | 0.060 | 1.642 | 1.874 |
|  | GP | -0.170 | 5.219 | 4.182 | 0.949 | 0.424 | -0.182 | 5.290 | 1.658 | 4.661 | 0.637 | 1.448 | 0.653 |
|  | CAST-D | 0.271 | 9.335 | 11.589 | 1.342 | 2.636 | 1.665 | 5.449 | 10.800 | 9.393 | 3.495 | 4.742 | 3.570 |
|  | CAST-L | 7.478 | 12.698 | 17.709 | 0.982 | 2.727 | 2.966 | 11.783 | 8.607 | 8.958 | 3.733 | 4.251 | 3.556 |

selection. Thus, we reserve 20% of the data for the test set and employ 3-fold cross-validation on the remainder to select the optimal hyperparameters. For the training dataset, 10% is randomly selected as the labeled data, with the remainder serving as unlabeled data for self-training. We compare the effect of diverse confidence within the self-training context using two primary self-training strategies: FPL and CPL. To determine the optimal $\alpha$ value for CAST, we execute a grid search in eight steps over the range [0.2, 0.75]. All experiments are conducted using ten random seeds ranging from 0 to 9, and the results are averaged across these runs. Further details regarding the datasets and implementations are provided in Appendix E.

### 4.2.2 RESULTS AND ANALYSIS.

**While calibrated confidences show little to no distinction compared to naive confidence, CAST significantly enhances confidence for self-training.**

Intuitively, reliable confidence in the self-training context should yield superior performance compared with naive confidence. However, as summarized in Table 1, self-training approaches based on calibrated confidence often do not lead to performance improvement, and at times even diminish the final performance compared to self-training with naive confidence. Contrarily, CAST consistently delivers notable enhancements in self-training across various strategies, datasets, and models. In all conducted experiments, CAST often outperforms the other approaches, securing the top position in every experiment and ranking second in most. We further investigate the effects of various

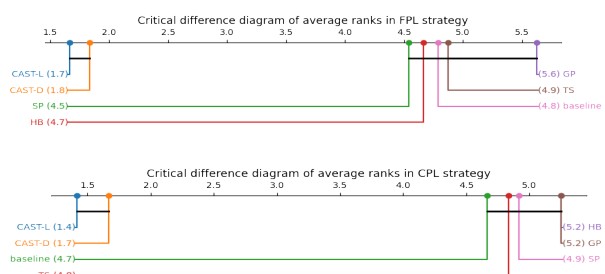

Figure 4: Critical difference diagrams of average ranks from Table 1 for FPL (Top) and for CPL (Bottom). Statistically equivalent methods are connected using horizontal bars.

confidences on self-training using a statistical approach, as shown in Figure 4. We employ the critical difference diagrams using average ranks of each confidence-based self-training for visualization, a standard visualizing method for statistical tests, as introduced by Demšar (2006). As depicted in Figure 4, regularized confidences differ substantially from naive confidence in the self-training context, whereas calibrated confidences do not. It verify that calibrating the confidence is meaningless in the context of self-training. Through our experiments and subsequent statistical analysis, it is evident that regularizing confidence to lower the confidence of pseudo-labels in low-density regions leads to performance gains in the self-training contexts. Conversely, confidence calibration does not yield such benefits. Appendix G provides the details of the statistical analysis.

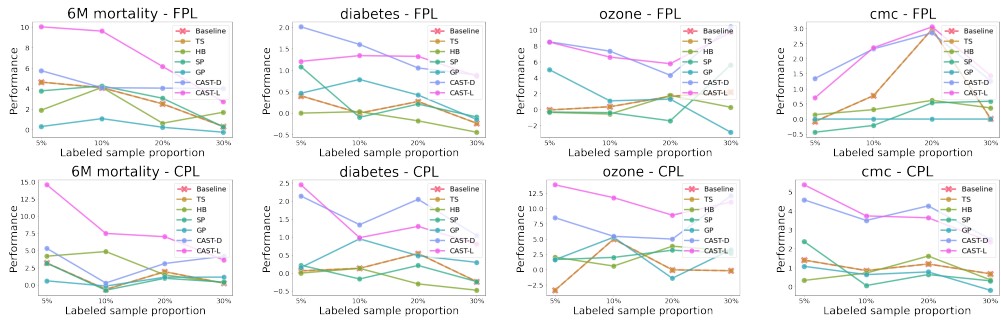

Figure 5: Relative improvement of various confidence-based self-training over various proportions of labeled samples in the training dataset.

**CAST demonstrates robustness for various labeled sample proportions.** Given that CAST derives prior knowledge from labeled data within the training dataset, we assess its effectiveness across various labeled sample proportions. We depict the outcomes of self-training using different confidences at labeled training sample proportions of {5%, 10%, 20%, 30%} with XGBoost across the four datasets in Figure 5. As illustrated in Figure 5, CAST consistently outperforms naive confidence-based self-training, irrespective of the labeled sample proportion in the training dataset. These findings underscore the robustness of CAST to variations in the proportion of labeled samples.

**CAST is robust to feature corruption.** Feature corruption is a common problem in many real-world scenarios. We investigate the effects of different confidences using XGBoost on datasets with corrupted features to demonstrate the robustness of CAST for noisy features. We outline the methodology for inducing feature corruption as follows. We randomly select a fraction of the features and replace each chosen feature with a value drawn from the empirical marginal distribution of that feature. This distribution is defined as a uniform distribution over the values that the feature takes on across the training dataset. The corruption ratio is fixed at 20% for each training sample. The results are summarized in Table 2. Clearly, CASTs consistently show notable performance improvements even in the presence of corrupted features.

Table 2: Relative improvement over four tabular datasets with corrupted features. The top results are highlighted in bold, while the second-best scores are underlined.

|  | FPL | | | | CPL | | | |
|---|---|---|---|---|---|---|---|---|
|  | 6M mortality | diabetes | ozone | cmc | 6M mortality | diabetes | ozone | cmc |
| Baseline | 6.519 | 0.000 | -3.013 | 0.680 | 5.353 | 0.153 | 9.181 | 1.105 |
| TS | 5.242 | 0.031 | **13.481** | 0.680 | 5.858 | -0.184 | 1.573 | 1.105 |
| HB | 6.963 | -0.367 | 10.235 | -0.551 | 6.030 | -0.337 | 1.931 | 0.509 |
| SP | 5.817 | -0.122 | -4.124 | 0.492 | 5.631 | 0.551 | 1.946 | 1.489 |
| GP | -1.922 | **1.836** | 5.087 | -1.437 | 2.411 | 0.061 | 15.127 | -0.531 |
| CAST-D | 8.280 | 1.499 | 12.432 | 1.188 | 9.181 | 1.285 | 16.634 | 3.864 |
| CAST-L | **12.902** | 1.714 | 8.508 | **1.312** | **12.152** | **2.295** | **24.050** | **4.188** |

### 4.2.3  HYPERPARAMETER $\alpha$

Here, we analyze the winning value of the hyperparameter $\alpha$ during the grid search for the experiments that are conducted for Table 1. Figure 6 depicts a plot summarizing the winning values of $\alpha$. The $\alpha$ is employed to determine the extent of the influence that prior knowledge on pseudo-label valuation in eq (4). Given that prior knowledge sourced from the training data distribution and the confidence of the classifier vary across datasets, models, and random seeds, a universal optimal value does not exist. However, we can recommend a search range for tuning the $\alpha$. We identify an upper bound of the 90% confidence interval for $\alpha$ as 0.7. Therefore, we suggest 0.7 or less when tuning the hyperparameter $\alpha$.

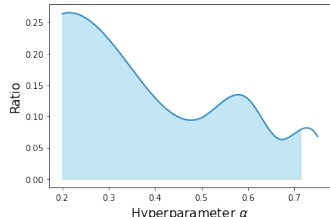

Figure 6: Plot of the winning values of the hyperparameter $\alpha$. The colored region denotes 90% of the confidence interval.

### 4.3 ADDITIONAL ANALYSIS FOR CAST

#### 4.3.1 COMBINATION OF CAST AND NOISE FILTERING

CAST is designed to be seamlessly integrated into existing self-training algorithms without requiring major alterations, making it a versatile add-on. This adaptability allows it to be paired with noise filtering techniques to achieve more reliable self-training. Table 3 shows the performance improvements when combining CAST with a Mahalanobis distance-based noise filtering approach, as employed by Tanha et al. (2017). Our experimental setup mirrors the one used in Section 4.2, except for the ozone dataset. This is because of the challenge of computing the Mahalanobis distance using only 10% of the labeled data of the ozone dataset. From the results in Table 3, it is clear that noise filtering with CAST provides a greater performance gain.

Table 3: Relative improvement of CAST with Mahalanobis distance-based noise filtering. The top results are highlighted in bold, while the second-best scores are underlined.

| | | 6M mortality | | | diabetes | | | cmc | | |
|---|---|---|---|---|---|---|---|---|---|---|
| | | XGB | FT | MLP | XGB | FT | MLP | XGB | FT | MLP |
| FPL | Baseline | 6.617 | 6.829 | 9.785 | 0.884 | 1.000 | 1.362 | 2.255 | 1.021 | 0.075 |
| | CAST-D | 5.970 | 8.741 | 14.777 | 3.470 | 1.970 | 2.452 | 3.178 | 2.895 | 2.754 |
| | CAST-L | 12.132 | 13.618 | 20.285 | 3.699 | 2.182 | 2.573 | 3.413 | 2.315 | 1.421 |
| CPL | Baseline | 4.297 | 8.449 | 9.685 | 1.669 | 2.030 | 2.149 | 4.065 | 4.787 | 2.911 |
| | CAST-D | 4.916 | 10.634 | 12.875 | 3.797 | 3.606 | 3.814 | 5.451 | 6.731 | 6.197 |
| | CAST-L | 12.953 | 15.080 | 21.104 | 3.273 | 3.636 | 4.177 | 5.549 | 6.905 | 5.355 |

#### 4.3.2 CAST CAN CHANGE THE MOST CONFIDENT CLASS

Unlike most previous methods regarding reliable pseudo-labeling (Li & Zhou, 2005; Rizve et al., 2021; Chen et al., 2022), CAST can change the most confident class. In essence, CAST regularizes the confidence of each pseudo-label based on class-specific prior knowledge. Consequently, the most confident class may change because the degree of regularization varies across the classes. We present results from naive self-training, which strictly determines pseudo-labels based on the most confident class, irrespective of the confidence magnitude (Lee et al., 2013). The results in Table 4 indicate that CAST can modify the most confident class to generate trustworthy pseudo-labels, thereby delivering superior performance over naive confidence-based self-training. Moreover, this capability explains the results in Section 4.3.1, as many noise filtering techniques identify noise based on the most confident class of unlabeled data.

Table 4: Relative improvement of naive self-training using different confidences. The top results are highlighted in bold, while the second-best scores are underlined.

| | 6M mortality | | | diabetes | | | ozone | | | cmc | | |
|---|---|---|---|---|---|---|---|---|---|---|---|---|
| | XGB | FT | MLP | XGB | FT | MLP | XGB | FT | MLP | XGB | FT | MLP |
| Baseline | 4.773 | 2.176 | 5.736 | 0.033 | 0.424 | 0.726 | 3.868 | 1.694 | 1.482 | 0.797 | 0.339 | 0.091 |
| CAST-D | 4.773 | 2.379 | 6.317 | 0.393 | 1.212 | 1.483 | 9.845 | 5.664 | 6.546 | 1.694 | 0.446 | 0.450 |
| CAST-L | 9.667 | 6.179 | 11.165 | 0.131 | 0.758 | 1.423 | 11.453 | 4.233 | 7.334 | 2.273 | 1.388 | 0.104 |

## 5 CONCLUSION

In this paper, we propose a novel self-training enhancing algorithm: CAST, which solely regularizes the confidence of the classifier to be aware of the cluster assumption and does not need any significant modification to the existing self-training algorithms or tabular models. Through extensive experiments across diverse settings, we verify that regularized confidence in CAST consistently improves self-training regardless of self-training strategies, datasets, and models, while calibrated confidence does not guarantee performance improvement in self-training. We additionally show some beneficial attributes of CAST and offer guidance on determining the search range for tuning hyperparameter $\alpha$. A current limitation of the CAST is its inapplicability to domains such as images or text as there are no suitable density estimation methods. For future work, we reserve direct assessments of confidence in the context of self-training without performing self-training iterations.

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

## A  EXPECTED CALIBRATION ERROR (ECE)

The Expected Calibration Error (ECE) (Naeini et al., 2015) quantifies the discrepancy between a model's predicted confidence and its true accuracy. To compute the ECE, predictions are grouped into $M$ bins of equal sizes based on their confidence, and the difference between the average accuracy and average confidence for each bin is determined.

Formally, the ECE is given by:

$$ECE = \sum_{m=1}^{M} \frac{|B_m|}{n} \left| acc(B_m) - conf(B_m) \right|, \tag{5}$$

where $B_m$ is the set of indices of samples whose prediction confidence falls within interval $I_m = (\frac{m-1}{M}, \frac{m}{M}]$, $|B_m|$ represents the number of predictions in the $m_{th}$ bin, $n$ denotes the total number of samples, and $acc$ and $conf$ denote the average accuracy and average confidence of each bin, respectively.

## B  WHY THE INHERENT VALUE OF CALIBRATION IN SELF-TRAINING REMAINS UNDEREXPLORED

Wang et al. (2021) demonstrate that graph neural networks (GNNs) tend to be under-confident and argue that the underperformance of existing self-training methods is caused by large numbers of high-accuracy predictions distributed in low-confidence intervals. Nevertheless, this under-confidence may not pose a significant hurdle for self-training strategies that employ a curriculum pseudo-labeling, which progressively reduces the threshold throughout self-training iterations to use the pseudo-labels that have low confidence. Furthermore, Munir et al. (2022) show that, although their calibration techniques effectively mitigate over-confidence issues after self-training for domain adaptive detectors, they do not consistently enhance average precision.

## C  WHY CAST IS EFFECTIVE?

Self-training is a version of the entropy minimization algorithm, which minimizes the likelihood deprived of the entropy of the partition (Amini & Gallinari, 2002). It constructs hard (one-hot) labels from high-confidence predictions on unlabeled data to implicitly achieve entropy minimization (Berthelot et al., 2019). The entropy minimization techniques assume that the cluster assumption is ensured in the dataset (Grandvalet & Bengio, 2004), and aim that the classifier learns the low-density separations in the data. However, unreliable pseudo-labels that lie in low-density regions, stemming from erroneous confidence, violate the assumption and consequently disrupt the classifier's ability to learn the separations among classes. On the other hand, CAST forces the pseudo-labels in low-density regions to have lower confidence to avoid the violation of the assumption. Therefore, CAST achieves more reliable pseudo-labels resulting in successful entropy minimization.

## D  IMPLEMENTATION DETAILS OF PRIOR KNOWLEDGE

In this section, we describe the implementations of our two density estimators to extract prior knowledge, multivariate kernel density estimator, and empirical likelihood.

### D.1  MULTIVARIATE KERNEL DENSITY ESTIMATOR

To estimate density to regularize the classifier's confidence, we employ a multivariate kernel density estimator provided by the `statsmodels` package (Seabold & Perktold, 2010). We follow the default kernel settings of `statsmodels`, which are the Gaussian kernel for continuous features and Aitchison-Aitken kernel for categorical features.

### D.2  EMPIRICAL LIKELIHOOD

The empirical likelihood does not require any assumption that the data come from a known family of distributions. Given the potential for many real-world datasets to be incomplete, distorted, or

subject to sampling bias, traditional density estimators might occasionally fall short in approximating true densities. Empirical likelihood, with its adaptability, has demonstrated effectiveness in such scenarios, as evidenced by numerous studies (Owen, 1988; 2001; Chen & Lazar, 2010). Therefore, we adopt empirical likelihood as another measure of density.

We implement a simplified variant of the empirical likelihood as follows: Let $\mathbf{x}^{(i)} = \left[x_1^{(i)}, x_2^{(i)}, ..., x_{m-1}^{(i)}, x_m^{(i)}\right]$ comprising $m$ features. Similarly, $\mathbf{y}^{(i)} = \left[y_1^{(i)}, y_2^{(i)}, ..., y_{N-1}^{(i)}, y_N^{(i)}\right]$ is a one-hot vector in the $N$-class dataset. If $y_j^{(i)} = 1$, the $i^{th}$ sample belongs to the $j^{th}$ class. For a given $\mathbf{x}^{(i)}$, the empirical likelihood of its pseudo-label $\tilde{y}_j^{(i)}$ is formulated as follows.

$$
\begin{aligned}
P(\mathbf{x}^{(i)}|\tilde{y}_j^{(i)} = 1) &= \frac{P(\tilde{y}_j^{(i)} = 1, x_1^{(i)}, x_2^{(i)}, ..., x_{m-1}^{(i)}, x_m^{(i)})}{P(\tilde{y}_j^{(i)} = 1)} \\
&= P(x_1^{(i)}|\tilde{y}_j^{(i)} = 1) \times P(x_2^{(i)}|\tilde{y}_j^{(i)} = 1) \times ... \times P(x_m^{(i)}|\tilde{y}_j^{(i)} = 1)
\end{aligned}
\tag{6}
$$

For simplicity, we operate under the premise that features of $\mathbf{x}$ are conditional independence, given the pseudo-label $\tilde{y}_j$. While conditional independence of feature is seldom a reality in many datasets, we are inspired by the assumption used in many successful studies that have used Naive Bayes [5]. Furthermore, we calculated the likelihood of pseudo-labels between selected features to alleviate the violation of the assumption, regarding the following hypothesis on which the heuristic is based Hall (2000)'s work: *Good feature subsets contain features highly correlated with the class, yet uncorrelated with each other* and some successful research to improve Naive Bayes using feature selection (Ratanamahatana & Gunopulos, 2003; Blanquero et al., 2021). Lastly, we use a log-likelihood by applying logarithms on eq (6) to enhance computational efficiency and prevent numerical errors.

For the categorical features, we determine the likelihood of each distinct value using their empirical distribution. For continuous features, we transform them into 10 discrete bins and subsequently calculate their likelihood based on the empirical distribution of these bins.

# E  EXPERIMENTAL DETAILS

## E.1  DATASET DETAILS

Table 5: Overview of datasets. We abbreviate "F1-score" as "F1," "balanced accuracy" as "b-acc," and "accuracy" as "acc".

| name | class | features | n_samples | metric | name | class | features | n_samples | metric |
|---|---|---|---|---|---|---|---|---|---|
| 6M mortality | 2 | 76 | 15628 | F1 | jm1 | 2 | 22 | 10855 | F1 |
| diabetes | 2 | 9 | 768 | acc | bioresponse | 2 | 1777 | 3751 | F1 |
| ozone (ozone-level-8hr) | 2 | 73 | 2534 | F1 | kc2 | 2 | 22 | 522 | F1 |
| cmc | 3 | 10 | 1473 | b-acc | kc1 | 2 | 22 | 2109 | F1 |
| kr-vs-kp | 2 | 37 | 3196 | acc | blood-transfusion-service-center | 2 | 5 | 748 | acc |
| credit-g | 2 | 21 | 1000 | b-acc | qsar-biodeg | 2 | 42 | 1055 | b-acc |
| sick | 2 | 30 | 3772 | f1 | wall-robot-navigation | 4 | 25 | 5456 | F1 |
| splice | 3 | 62 | 3190 | b-acc | churn | 2 | 21 | 5000 | F1 |
| vehicle | 4 | 19 | 846 | acc | car | 4 | 7 | 1728 | b-acc |
| pc4 | 2 | 38 | 1458 | F1 | steel-plates-fault | 7 | 28 | 1941 | F1 |
| pc3 | 2 | 38 | 1563 | F1 | | | | | |

**Dataset preprocessing.** We use label encoding for all categorical features, except for the 6M mortality dataset where certain categorical features necessitate one-hot encoding. We impute missing data using an iterative imputer in scikit-learn package (Pedregosa et al., 2011). For MLP, we embed categorical features in high-dimensional spaces and apply batch normalization to the continuous features.

---

[5]For example, even with correlated features, Naive Bayes, which operates under the conditional independence assumption, often has produced commendable results on a variety of tabular datasets (Hand & Yu, 2001)

### E.2 IMPLEMENTATION DETAILS OF TABULAR MODELS

We use Pytorch Tabular framework for FT-Transformer and MLP (Joseph, 2021), and the official Python package for XGBoost.

### E.3 DETAILS OF CONFIDENCE CALIBRATION

We use netcal framework (Küppers et al., 2020) for temperature scaling and histogram binning. We adopt six knots for the spline calibration according to the Gupta et al. (2020)'s work and do not use any hyperparameters for the latent Gaussian process calibration since it is a nonparametric method. Then, we fit calibration methods to the validation dataset except for spline calibration which does not require the fitting procedure.

### E.4 DETAILS OF HYPERPARAMETER TUNNING

Table 6: Optuna hyperparameter search space for XGBoost

| Hyperparameter | Search Method | Search Space |
| --- | --- | --- |
| max_leaves | suggest_int | [300,4000] |
| n_estimators | suggest_int | [10,3000] |
| learning_rate | suggest_uniform | [0,1] |
| max_depth | suggest_int | [3, 20] |
| scale_pos_weight | suggest_int | [1, 100] |

Table 7: Optuna hyperparameter search space for FT-Transformer

| Hyperparameter | Search Method | Search Space |
| --- | --- | --- |
| input_embed_dim | suggest_categorical | [16,24,32,48] |
| embedding_dropout | suggest_uniform | [0.05,0.3] |
| share_embedding | suggest_categorical | [True, False] |
| num_heads | suggest_categorical | [1,2,4,8] |
| num_attn_blocks | suggest_int | [2,10] |
| transformer_activation | suggest_categorical | [GEGLU, ReGLU, SwiGLU] |
| use_batch_norm | suggest_categorical | [True, False] |
| batch_norm_continuous_input | suggest_categorical | [True, False] |
| learning_rate | suggest_uniform | [0.0001, 0.05] |
| scheduler_gamma | suggest_uniform | [0.1, 0.95] |
| scheduler_step_size | suggest_int | [10, 100] |

Table 8: Optuna hyperparameter search space for MLP

| Hyperparameter | Search Method | Search Space |
| --- | --- | --- |
| embedding_dropout | suggest_uniform | [0, 0.2] |
| layers | suggest_categorical | [128-64-32, 256-128-64, 128-64-32-16, 256-128-64-32] |
| activation | suggest_categorical | [ReLU, LeakyReLU] |
| learning_rate | suggest_uniform | [0.0001, 0.05] |
| scheduler_gamma | suggest_uniform | [0.1, 0.95] |
| scheduler_step_size | suggest_int | [10, 100] |

## F    ADDITIONAL EXPERIMENT WITH TOY DATASET

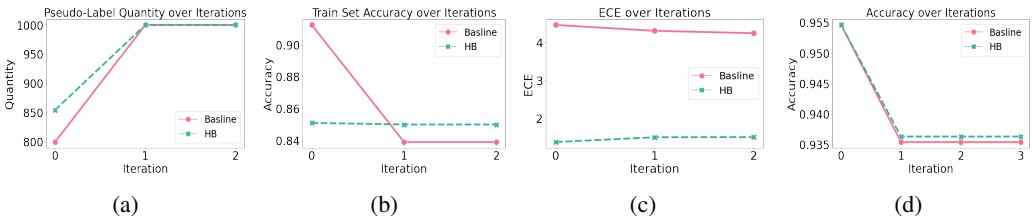

|     (a)     |     (b)     |     (c)     |     (d)     |

Figure 7: Pseudo-label quantity, training set accuracy, test ECE, and test accuracy, in sequence, for each confidence-based self-training over iterations.

We conduct an additional experiment using a toy dataset to show the results with a high threshold. Except for the threshold, we use the same setup described in Section 4.1. We set a threshold ($\tau$) of 0.9 for FPL and limit our comparison to naive confidence and HB, as the high threshold is not compatible with regularized confidence. In Figure 7, it is evident that although we adopt a high threshold for FPL, naive confidence, and calibrated confidence-based self-training fail to generate reliable pseudo-labels and improve performance.

## G    STATISTICAL ANALYSIS FOR EMPIRICAL RESULTS

We conduct Friedman test and the results in Table 9 show that we can confidently reject the null hypothesis given the considerably small p-value. Therefore, we conduct a Conover post-hoc test and visualize the results using the critical difference diagrams shown in Figure 4. We use a significance level $\alpha = 0.05$ for our critical difference diagram. We also conduct Holm's test as an additional post-hoc test, and report the results between naive confidence and the others in Table 9. This also proves that CASTs are significantly different from naive confidence-based self-training.

Table 9: Statistic Analyze

|  | FPL | | CPL | |
|---|---|---|---|---|
|  | statistic | p-value | statistic | p-value |
| Friedman | 40.1455 | 4.26e-07 | 45.9213 | 3.06e-08 |
|  | Holm's test with adjusted $\alpha$'s (0.05) | | | |
|  | p-value | | p-value | |
| TS | 1.0000 | | 1.0000 | |
| HB | 1.0000 | | 1.0000 | |
| SP | 1.0000 | | 1.0000 | |
| GP | 0.3760 | | 0.8486 | |
| CAST-D | **0.0103** | | **0.0103** | |
| CAST-L | **0.0103** | | **0.0103** | |

## H   COMPUTATIONAL COST OF CAST

We have presented the computational cost of CAST in Table 10. The results display the mean of ten self-training experiments using a curriculum-based pseudo-labeling approach. The experiments were carried out using one CPU core of Ryzen 5975wx and one RTX 4090. As seen in the table, the computational costs of CAST-D and CAST-L are almost as close as the training time of XGBoost. In particular, the cost is significantly lower than the training time of neural networks.

Table 10: Additional computational cost of CAST.

| | Time (s) | | | | | | | | | | | |
| | 6M mortality | | | diabetes | | | ozone | | | cmc | | |
| | XGB | FT | MLP | XGB | FT | MLP | XGB | FT | MLP | XGB | FT | MLP |
|---|---|---|---|---|---|---|---|---|---|---|---|---|
| Training | 5.25 | 749.02 | 96.97 | 0.10 | 26.85 | 9.40 | 0.51 | 89.57 | 14.32 | 0.19 | 27.90 | 16.12 |
| CAST-D | | 7.13 | | | 0.15 | | | 0.50 | | | 0.25 | |
| CAST-L | | 3.99 | | | 0.14 | | | 0.48 | | | 0.19 | |
| | Relative additional overhead of CAST compared to training time (%) | | | | | | | | | | | |
| | 6M mortality | | | diabetes | | | ozone | | | cmc | | |
| | XGB | FT | MLP | XGB | FT | MLP | XGB | FT | MLP | XGB | FT | MLP |
| CAST-D | 135.84 | 0.95 | 7.35 | 148.15 | 0.57 | 1.63 | 98.72 | 0.56 | 3.51 | 127.70 | 0.89 | 1.53 |
| CAST-L | 76.08 | 0.53 | 4.12 | 133.65 | 0.52 | 1.47 | 94.99 | 0.54 | 3.38 | 97.35 | 0.68 | 1.17 |

## I   ADDITIONAL EMPIRICAL RESULTS

Table 11: Relative improvement over seventeen tabular datasets. The top results are highlighted in bold, while the second-best scores are underlined.

| | | kr-vs-kp | credit-g | sick | splice | vehicle | pc4 | pc3 | jm1 | bioresponse |
|---|---|---|---|---|---|---|---|---|---|---|
| FPL | Baseline | 0.132 | 0.034 | 0.128 | 1.092 | 0.413 | 10.254 | 5.458 | 3.087 | -0.024 |
| | TS | 0.022 | -0.021 | **1.785** | 1.092 | 0.413 | 15.913 | 8.737 | 0.481 | 0.052 |
| | HB | 0.094 | -0.481 | 1.108 | 1.027 | 0.550 | 3.335 | 1.954 | 0.000 | 0.022 |
| | SP | 0.237 | 0.021 | 0.156 | 1.067 | 0.309 | 6.718 | 1.238 | 1.755 | -0.058 |
| | GP | **0.584** | -0.440 | 1.198 | -0.139 | 0.000 | 5.143 | 1.002 | 0.802 | 0.567 |
| | CAST-D | 0.485 | 1.745 | 0.000 | 0.503 | 1.788 | 20.554 | 11.785 | **5.134** | 0.688 |
| | CAST-L | 0.457 | **1.759** | 1.417 | **1.152** | **2.475** | **21.785** | **11.901** | 3.785 | **1.150** |
| CPL | Baseline | 0.336 | 0.900 | 1.477 | 0.400 | 0.584 | 7.438 | 7.310 | 5.113 | 0.155 |
| | TS | 0.413 | -0.412 | -0.073 | 0.349 | 0.584 | 18.044 | 4.094 | 3.311 | 0.290 |
| | HB | 0.397 | -0.124 | 0.475 | **0.837** | 1.719 | 15.031 | 2.929 | 0.497 | 0.354 |
| | SP | 0.386 | 0.323 | 1.668 | 0.469 | 0.069 | 6.232 | 6.239 | 4.222 | 0.265 |
| | GP | 0.595 | -1.141 | 1.513 | 0.544 | **4.056** | 5.772 | 2.498 | 4.594 | 0.249 |
| | CAST-D | 0.689 | 2.693 | **2.447** | 0.605 | 0.894 | **25.817** | 13.426 | **5.177** | **2.454** |
| | CAST-L | **0.782** | **3.731** | 1.508 | 0.626 | 3.025 | 22.177 | **16.176** | 4.828 | 1.821 |

| | | kc2 | kc1 | blood | qsar-biodeg | robot | churn | car | steel | Avg Rank (std) |
|---|---|---|---|---|---|---|---|---|---|---|
| FPL | Baseline | 1.792 | 0.870 | 0.548 | -0.045 | 0.095 | 3.267 | 0.731 | -0.203 | 4.38(1.16) |
| | TS | 0.359 | 3.203 | 0.061 | 0.414 | 0.095 | 0.915 | 0.731 | -0.203 | 4.44(1.59) |
| | HB | 0.283 | 0.000 | 0.000 | 0.937 | 0.114 | 1.129 | 0.066 | 0.456 | 5.12(1.49) |
| | SP | 1.767 | -1.773 | 1.552 | -0.269 | 0.162 | 3.407 | -0.025 | -0.089 | 5.00(1.37) |
| | GP | -0.135 | 0.298 | 1.765 | -0.155 | 0.000 | 0.682 | 0.020 | 0.000 | 5.26(1.83) |
| | CAST-D | **5.432** | **7.470** | **1.856** | **1.583** | **0.355** | 5.878 | **4.492** | 0.000 | 2.21(1.71) |
| | CAST-L | 2.984 | 6.701 | 1.826 | 1.200 | 0.304 | **5.929** | 3.007 | **1.394** | **1.59(0.60)** |
| CPL | Baseline | 2.662 | 4.862 | 1.704 | -0.180 | 0.067 | 4.673 | 2.903 | 0.038 | 4.76(1.68) |
| | TS | 0.648 | 3.591 | -0.061 | 0.808 | 0.067 | 1.264 | 2.903 | 0.038 | 5.47 (1.40) |
| | HB | 3.326 | 2.008 | 1.339 | 1.312 | 0.257 | 1.657 | 2.381 | 0.342 | 4.53(1.75) |
| | SP | 2.557 | 2.940 | 1.339 | -0.132 | 0.076 | 4.017 | 2.809 | 0.177 | 5.00(1.08) |
| | GP | 2.473 | 3.485 | 1.765 | 0.244 | 0.371 | 5.180 | 1.379 | 0.634 | 4.41(1.94) |
| | CAST-D | **9.079** | **6.714** | **2.830** | 1.193 | **0.377** | **8.868** | **5.405** | 0.076 | **1.82(1.20)** |
| | CAST-L | 3.500 | 4.704 | 2.161 | **1.510** | 0.336 | 7.982 | 4.022 | **1.495** | 2.00(0.84) |

## J  ABSOLUTE PERFORMANCE

Table 12: Absolute performance over four tabular datasets. The top results are highlighted in bold, while the second-best scores are underlined.

| | | 6M mortality | | | diabetes | | | ozone | | | cmc | | |
|---|---|---|---|---|---|---|---|---|---|---|---|---|---|
| | | XGB | FT | MLP | XGB | FT | MLP | XGB | FT | MLP | XGB | FT | MLP |
| Supervised Learning | | 0.4055 | 0.3806 | 0.3311 | 0.6613 | 0.7143 | 0.7152 | 0.2803 | 0.3769 | 0.3823 | 0.4638 | 0.4696 | 0.4437 |
| FPL | Baseline | 0.4221 | 0.3849 | 0.3571 | 0.6613 | 0.7167 | 0.7245 | 0.2813 | 0.3781 | 0.3872 | 0.4674 | 0.4708 | 0.4443 |
| | TS | 0.4221 | 0.3849 | 0.3566 | 0.6613 | 0.7167 | 0.7245 | 0.2813 | 0.3782 | 0.3872 | 0.4674 | 0.4708 | 0.4443 |
| | HB | 0.4222 | 0.3806 | 0.3306 | 0.6615 | 0.7214 | 0.7208 | 0.2787 | 0.3864 | 0.3817 | 0.4652 | 0.4745 | 0.4481 |
| | SP | 0.4228 | 0.3895 | 0.3590 | 0.6606 | 0.7158 | 0.7214 | 0.2793 | 0.3958 | 0.3921 | 0.4628 | 0.4762 | 0.4437 |
| | GP | 0.4099 | 0.3764 | 0.3308 | 0.6665 | 0.7199 | 0.7136 | 0.2834 | 0.3807 | 0.3768 | 0.4638 | 0.4696 | 0.4437 |
| | CAST-D | 0.4221 | 0.4018 | 0.3660 | **0.6719** | **0.7242** | 0.7275 | **0.3009** | **0.4103** | **0.4169** | 0.4746 | 0.4730 | **0.4509** |
| | CAST-L | **0.4444** | **0.4147** | **0.3873** | 0.6701 | 0.7190 | **0.7292** | 0.2988 | 0.4022 | 0.4131 | **0.4747** | **0.4780** | 0.4483 |
| CPL | Baseline | 0.4029 | 0.4039 | 0.3471 | 0.6621 | 0.7201 | 0.7208 | 0.2943 | 0.3683 | 0.4047 | 0.4677 | 0.4786 | 0.4456 |
| | TS | 0.4029 | 0.4039 | 0.3471 | 0.6621 | 0.7201 | 0.7208 | 0.2943 | 0.3683 | 0.4038 | 0.4677 | 0.4786 | 0.4456 |
| | HB | 0.4252 | 0.3893 | 0.3536 | 0.6621 | 0.7260 | 0.7177 | 0.2820 | 0.3845 | 0.3840 | 0.4671 | 0.4751 | 0.4447 |
| | SP | 0.4024 | 0.4047 | 0.3586 | 0.6602 | 0.7245 | 0.7180 | 0.2859 | 0.3817 | 0.4041 | 0.4641 | 0.4773 | 0.4520 |
| | GP | 0.4048 | 0.4005 | 0.3449 | 0.6675 | 0.7173 | 0.7139 | 0.2952 | 0.3831 | 0.4001 | 0.4667 | 0.4764 | 0.4466 |
| | CAST-D | 0.4066 | 0.4162 | 0.3694 | **0.6701** | 0.7331 | 0.7271 | 0.2956 | **0.4176** | **0.4182** | 0.4800 | **0.4919** | **0.4595** |
| | CAST-L | **0.4359** | **0.4290** | **0.3897** | 0.6677 | **0.7338** | **0.7364** | **0.3133** | 0.4093 | 0.4165 | **0.4811** | 0.4896 | 0.4595 |

Table 13: Absolute performance over seventeen tabular datasets. The top results are highlighted in bold, while the second-best scores are underlined.

| | | kr-vs-kp | credit-g | sick | splice | vehicle | pc4 | pc3 | jm1 | bioresponse |
|---|---|---|---|---|---|---|---|---|---|---|
| Supervised Learning | | 0.9453 | 0.5775 | 0.7473 | 0.9077 | 0.5704 | 0.2467 | 0.2512 | 0.3694 | 0.7436 |
| FPL | Baseline | 0.9466 | 0.5777 | 0.7482 | 0.9176 | 0.5727 | 0.2719 | 0.2649 | 0.3808 | 0.7434 |
| | TS | 0.9455 | 0.5774 | **0.7606** | 0.9176 | 0.5727 | 0.2859 | 0.2731 | 0.3712 | 0.7440 |
| | HB | 0.9462 | 0.5748 | 0.7556 | 0.9170 | 0.5735 | 0.2549 | 0.2561 | 0.3694 | 0.7438 |
| | SP | 0.9476 | 0.5777 | 0.7484 | 0.9174 | 0.5722 | 0.2632 | 0.2543 | 0.3759 | 0.7432 |
| | GP | **0.9508** | 0.5750 | 0.7562 | 0.9064 | 0.5704 | 0.2593 | 0.2537 | 0.3724 | 0.7478 |
| | CAST-D | 0.9499 | 0.5876 | 0.7473 | 0.9122 | 0.5806 | 0.2974 | 0.2808 | **0.3884** | 0.7487 |
| | CAST-L | 0.9496 | **0.5877** | 0.7579 | **0.9181** | 0.5845 | 0.3004 | 0.2811 | 0.3834 | **0.7522** |
| CPL | Baseline | 0.9485 | 0.5827 | 0.7583 | 0.9113 | 0.5737 | 0.2650 | 0.2695 | 0.3883 | 0.7448 |
| | TS | 0.9492 | 0.5752 | 0.7467 | 0.9108 | 0.5737 | 0.2912 | 0.2615 | 0.3816 | 0.7458 |
| | HB | 0.9491 | 0.5768 | 0.7508 | **0.9153** | 0.5802 | 0.2837 | 0.2585 | 0.3712 | 0.7462 |
| | SP | 0.9490 | 0.5794 | 0.7597 | 0.9119 | 0.5708 | 0.2620 | 0.2668 | 0.3850 | 0.7456 |
| | GP | 0.9509 | 0.5710 | 0.7586 | 0.9126 | **0.5935** | 0.2609 | 0.2574 | 0.3864 | 0.7455 |
| | CAST-D | 0.9518 | 0.5931 | **0.7656** | 0.9132 | 0.5755 | **0.3103** | 0.2849 | **0.3885** | **0.7619** |
| | CAST-L | **0.9527** | **0.5991** | 0.7585 | 0.9133 | 0.5876 | 0.3014 | **0.2918** | 0.3872 | 0.7572 |

| | | kc2 | kc1 | blood | qsar-biodeg | robot | churn | car | steel | Avg Rank (std) |
|---|---|---|---|---|---|---|---|---|---|---|
| Supervised Learning | | 0.5091 | 0.3844 | 0.7302 | 0.7291 | 0.9624 | 0.5859 | 0.5472 | 0.6762 | - |
| FPL | Baseline | 0.5182 | 0.3877 | 0.7342 | 0.7287 | 0.9633 | 0.6051 | 0.5512 | 0.6748 | 4.38(1.16) |
| | TS | 0.5109 | 0.3967 | 0.7307 | 0.7321 | 0.9633 | 0.5913 | 0.5512 | 0.6748 | 4.44(1.59) |
| | HB | 0.5105 | 0.3844 | 0.7302 | 0.7359 | 0.9635 | 0.5926 | 0.5476 | 0.6793 | 5.12(1.49) |
| | SP | 0.5181 | 0.3776 | 0.7416 | 0.7271 | 0.9639 | 0.6059 | 0.5471 | 0.6756 | 5.00(1.37) |
| | GP | 0.5084 | 0.3855 | 0.7431 | 0.7279 | 0.9624 | 0.5899 | 0.5474 | 0.6762 | 5.26(1.83) |
| | CAST-D | **0.5367** | **0.4131** | **0.7438** | 0.7406 | **0.9658** | 0.6204 | **0.5718** | 0.6762 | 2.21(1.71) |
| | CAST-L | 0.5243 | 0.4102 | 0.7436 | 0.7378 | 0.9653 | **0.6207** | 0.5637 | **0.6856** | **1.59(0.60)** |
| CPL | Baseline | 0.5226 | 0.4031 | 0.7427 | 0.7278 | 0.9630 | 0.6133 | 0.5631 | 0.6764 | 4.76(1.68) |
| | TS | 0.5124 | 0.3982 | 0.7298 | 0.7350 | 0.9630 | 0.5934 | 0.5631 | 0.6764 | 5.47 (1.40) |
| | HB | 0.5260 | 0.3921 | 0.7400 | 0.7386 | 0.9648 | 0.5957 | 0.5603 | 0.6785 | 4.53(1.75) |
| | SP | 0.5221 | 0.3957 | 0.7400 | 0.7281 | 0.9631 | 0.6095 | 0.5626 | 0.6774 | 5.00(1.08) |
| | GP | 0.5217 | 0.3978 | 0.7431 | 0.7309 | 0.9659 | 0.6163 | 0.5548 | 0.6805 | 4.41(1.94) |
| | CAST-D | **0.5553** | **0.4102** | **0.7509** | 0.7378 | **0.9660** | **0.6379** | **0.5768** | 0.6767 | **1.82(1.20)** |
| | CAST-L | 0.5269 | 0.4025 | 0.7460 | **0.7401** | 0.9656 | 0.6327 | 0.5693 | **0.6863** | 2.00(0.84) |

