# OpenReview forum: "CAST: Cluster-Aware Self-Training for Tabular Data"
_ICLR.cc/2024/Conference — Submitted to ICLR 2024_

### Official Review · Reviewer_b9QV · 2023-10-21

**Soundness:** 3 good
**Presentation:** 3 good
**Contribution:** 3 good
**Rating:** 6
**Confidence:** 4

**Summary:**

This paper proposed a simple but effective self-training method for tabular data, which takes the cluster assumption to regularize confidence values. Experiments on four datasets demonstrate the superiority of the proposed method.

**Strengths:**

1. The proposed method is well motivated by the observations shown in Fig. 1, namely, pseudo-labels that lie in high-density regions are more reliable than those that lie in low-density regions.
2. The paper is well-written and organized in general. The simple modifications on the confidence value proved to be effective through experiments on four tabular datasets.

**Weaknesses:**

1. The density estimation plays an important role in the proposed method. However, the authors only spend a few words saying that the density is estimated using the prior knowledge derived from the labeled training data distribution. I am confused when reading this part of the method and hope the authors can provide more details on that.
2. Just as the authors have claimed, the only difference between CAST and the conventional self-training algorithm is the use of regularized confidence. In other words, it seems that the proposed method has no specific designs for tabular data. Thus, I wonder if it is possible to supply a bit more results on other forms of data to show the proposed method is a general solution in self-training.
3. Are the best choices of the hyper-parameter $\alpha$ the same across different datasets? Would the optimal value be influenced by the number of samples in the dataset?
4. There are some related self-training enhanced clustering methods such as SCAN (ECCV 2020), SPICE (TIP 2022), and TCL (IJCV 2022), that the authors are encouraged to include in the related works.
5. The meaning of the abbreviation could be provided in the caption of Table 1 to improve readability.

**Questions:**

Please refer to the weaknesses.

---

> ### Author Response · Authors · 2023-11-16
>
> We deeply appreciate your constructive feedback on our manuscript. Your insights have helped us refine our work, and we have updated the manuscript accordingly.
>
> ---
>
> ### Weakness 1
> The density estimation plays an important role in the proposed method. However, the authors only spend a few words saying that the density is estimated using the prior knowledge derived from the labeled training data distribution. I am confused when reading this part of the method and hope the authors can provide more details on that.
>
> *Answer:* We apologize for any confusion regarding the density estimation method in our manuscript. To clarify, we have revised the method section to detail the process as follows.
>
> We get the estimated density for unlabeled samples by extracting the prior knowledge using a density estimator $\textit{D}_{t}$ (e.g. multivariate kernel density estimator or empirical likelihood) which is fitted to the labeled training data distribution
> Here, the prior knowledge $\boldsymbol{\gamma}$ for each class is defined as follows:
>
> $$
> \boldsymbol{\gamma}^{(i)} \leftarrow \textit{D}_{t}(\boldsymbol{x}^{(i)}), \quad \text{where} \quad \boldsymbol{\gamma}^{(i)} = \text{[}{\gamma}_1, \gamma_2, ... , \gamma_N\text{]}
> $$
>
> Then, we normalize $\boldsymbol{\gamma}$ using a min-max scaler because the scale of $\boldsymbol{\gamma}$ varies among implementations, and we need a relative measure to align unlabeled samples.
>
> And we leave the specific choice of density estimator for CAST to implementation as we believe open to extension is a crucial component for the versatility. For example, CAST which uses dataset-specific density estimator might be surpass CAST-D or CAST-L on the specific dataset. Our implementation of CAST-D and CAST-L are shown in Appendix D.
>
>
> ### Weakness 2
> Just as the authors have claimed, the only difference between CAST and the conventional self-training algorithm is the use of regularized confidence. In other words, it seems that the proposed method has no specific designs for tabular data. Thus, I wonder if it is possible to supply a bit more results on other forms of data to show the proposed method is a general solution in self-training.
>
> *Answer:* We agree with your observation that the core idea of CAST, which involves adjusting the prediction confidence based on the cluster assumption, could extend to data types beyond tabular data. However, as we now note in our conclusion, the current implementation of CAST is limited by the absence of suitable density estimation methods for non-tabular data types, such as images or text. The structured nature of tabular data allows for clear density estimation from labeled training datasets as tabular data ensures that each feature occupies a specific, fixed position within the table. But there is no such characteristics in other data types. This limitation is now explicitly acknowledged in our manuscript.
>
> ### Weakness 3
> Are the best choices of the hyper-parameter $\alpha$ the same across different datasets? Would the optimal value be influenced by the number of samples in the dataset?
>
> *Answer:* As indicated in Figure 6 of our manuscript, there is no universally optimal choice for the hyperparameter $\alpha$. Its optimal value varies based on various factors, even within the same dataset, such as the number of training samples, classifier architecture, and self-training algorithms. Therefore, we recommend a specific range for tuning $\alpha$ to accommodate these variations.
>
>
> ### Weakness 4
> There are some related self-training enhanced clustering methods such as SCAN (ECCV 2020), SPICE (TIP 2022), and TCL (IJCV 2022), that the authors are encouraged to include in the related works.
>
> *Answer:* After reading the suggested literature, we have included SPICE (TIP 2022) and TCL (IJCV 2022) in the related works section as follows: Niu et al. (2022) ensure the reliability of pseudo-labels through the use of a semantically consistent ratio, while Li et al. (2022) enhance clustering performance by selectively incorporating the most confident predictions from each cluster. We also agree that SCAN (ECCV 2020) is significant in semi- and self-supervised learning, but their work does not improve self-training. Therefore, we exclude it from our related works section as it does not align directly with our work on enhanced self-training.
>
>
> ### Weakness 5
> The meaning of the abbreviation could be provided in the caption of Table 1 to improve readability.
>
> *Answer:* We have included detailed explanations of the abbreviations in the caption of Table 1.
>
> ---
>
> We hope these revisions adequately address your concerns and further clarify our research contributions. We are grateful for your valuable insights and remain open to any further feedback.

---

### Official Review · Reviewer_QLpL · 2023-10-27

**Soundness:** 2 fair
**Presentation:** 3 good
**Contribution:** 2 fair
**Rating:** 5
**Confidence:** 3

**Summary:**

The paper proposes a new calibration strategy for self-training on tabular data. The strategy is based on an algorithm outputting a confidence score for each input sample that linearly interpolates between the confidence score provided by the classifier and its scaled version. Specifically, the scaling factor incorporated the low-density assumption, thus being proportional to the data density. It is estimated either using a kernel density estimator or a Naive Bayes-like generative model. The overall calibration strategy can be easily plugged into existing self-training algorithms. Experiments are conducted on different toy and tabular datasets showcasing (i) the versatility of the approach for being easily applied to different self-training variants (fixed/adaptive threshold, noise filtering) and different classifiers (decision trees, MLPs) and (ii) the superiority against basic calibration strategies.

**Strengths:**

1. The idea is simple, yet novel (**Novelty**)
2. The paper is clear and easy to read (**Clarity**)
3. Code is provided. However, no additional check on replicability has been performed (**Reproducibility**)

**Weaknesses:**

1. The scope of applicability of the proposed solution is quite narrow. Indeed, the proposed solution seems to be applicable to low dimensional datasets and it is not clear how well the solution scales and generalises to more realistic high-dimensional datasets (**Significance**)
2. The proposed solution requires a density estimation step and therefore it is more computationally demanding with respect to the considered baselines. Experiments should provide also this information (**Quality**)
3. The cluster assumption (or equivalently the low-density separation) can be cheaply incorporated by leveraging techniques based on entropy minimisation for semi-supervised learning. A discussion and possibly experimental comparison against such techniques is missing (**Quality**). For instance, see [1-3]
4. Limitations are not discussed (**Quality**)

**References**

[1] Semi-supervised Learning by Entropy Minimisation. NeurIPS 2004

[2] Towards making unlabeled data never hurt. PAMI 2015

[3] MixMatch: A Holistic Approach to Semi-Supervised Learning. NeurIPS 2019

**Questions:**

Please find below some questions related to the above mentioned weaknesses plus some more detailed ones about the experiments:
1. Can you please elaborate on the 4 above-mentioned weaknesses?
2. Regarding experiments on toy datasets, is there any reason why temperature scaling is not shown?
3. In almost all experiments there is a significant difference between the two proposed ways of estimating the density (CAST-D and CAST-L). Can you please discuss about this aspect? Is this issue related to an improper hyperparameter tuning?
4. In Figure 5, can you please explain why the performance decrease with a larger amount of labeled examples, as this seems a counterintuitive result?

---

> ### Author Response · Authors · 2023-11-16
> **Official Comment by Authors (1)**
>
> We are grateful for your insightful feedback and the opportunity to further clarify and enhance our manuscript.
>
> ---
>
> ### Weakness 1
> The scope of applicability of the proposed solution is quite narrow. Indeed, the proposed solution seems to be applicable to low dimensional datasets and it is not clear how well the solution scales and generalizes to more realistic high-dimensional datasets *(Significance)*
>
> *Answer:* We understand the concern about CAST's scalability to high-dimensional datasets. To address this, we included results from the Bioresponse dataset in OpenML-CC18, which is notably high-dimensional with 1777 features and 3751 samples. The results, detailed in Table 11 in Appendix I, demonstrate that CAST effectively handles high-dimensional data, showing significant improvements of CAST.
>
> | | Method  | Relative Improvement | | Method  | Relative Improvement |
> |---------|---------|----------|---------|---------|----------|
> | | Baseline| -0.024    | | Baseline| 0.155   |
> | | TS      | 0.052   | | TS      | 0.290  |
> | | HB      | 0.022    | | HB      | 0.354    |
> FPL| SP    | -0.058    |CPL| SP    | 0.265    |
> | | GP      | 0.567    | | GP      | 0.249    |
> | | CAST-D  | 0.688    | | CAST-D  | 2.454    |
> | | CAST-L  | 1.150    | | CAST-L  | 1.821    |
>
> ### Weakness 2
> The proposed solution requires a density estimation step and therefore it is more computationally demanding with respect to the considered baselines. Experiments should provide also this information (*Quality*)
>
> *Answer:* We have added detailed information about CAST’s computational cost in Appendix H. The computational costs for CAST-D and CAST-L were benchmarked against the training time of XGBoost, using a single CPU core of Ryzen 5975wx and one RTX 4090. The results indicate that CAST's computational demand is comparable to the training time of XGBoost and significantly lower than that of neural networks.
>
> |            |            |            |           |         |         |    Time (s)  |       |         |          |           |          |            |
> |------------|--------|--------|-------|-------|-------|-------|-------|-------|-------|-------|-------|-------|
> |            |            | 6M mortality | |          | diabetes |       |          | ozone |           |           | cmc  |            |
> |            | XGB    | FT     | MLP   | XGB   | FT    | MLP   | XGB   | FT    | MLP   | XGB   | FT    | MLP   |
> | Training   | 5.25   | 749.02 | 96.97 | 0.10  | 26.85 | 9.40  | 0.51  | 89.57 | 14.32 | 0.19  | 27.90 | 16.12 |
> | CAST-D     |    | 7.13      |      |   | 0.15     |      |   | 0.50    |      |  | 0.25      |      |
> | CAST-L     |  | 3.99      |     |  | 0.14      |      |   | 0.48     |      |   | 0.19     |     |
> |            |            |            |       Relative    |      additional   |     overhead    |       of CAST     |  compared     |     to training    |   time (%)        |           |          |            |
> |            |            | 6M mortality | |          | diabetes |       |          | ozone |           |           | cmc  |            |
> |            | XGB    | FT     | MLP   | XGB   | FT    | MLP   | XGB   | FT    | MLP   | XGB   | FT    | MLP   |
> | CAST-D   | 135.84 | 0.95   | 7.35  | 148.15| 0.57  | 1.63  | 98.72 | 0.56  | 3.51  | 127.70| 0.89  | 1.53  |
> | CAST-L   | 76.08  | 0.53   | 4.12  | 133.65| 0.52  | 1.47  | 94.99 | 0.54  | 3.38  | 97.35 | 0.68  | 1.17  |
>
> ### Weakness 3
> The cluster assumption (or equivalently the low-density separation) can be cheaply incorporated by leveraging techniques based on entropy minimisation for semi-supervised learning. A discussion and possibly experimental comparison against such techniques is missing (*Quality*). For instance, see [1-3]
>
> *Answer:* Self-training is a version of the entropy minimization algorithm, which minimizes the likelihood deprived of the entropy of the partition [1]. It constructs hard (one-hot) labels from high-confidence predictions on unlabeled data to implicitly achieve entropy minimization [2]. We have clarified that self-training is inherently an entropy minimization algorithm. And we acknowledge the significance of entropy minimization techniques in semi-supervised learning. However, we intentionally did not compare CAST with other SSL techniques including other entropy minimization algorithms as our aim is to demonstrate whether self-training can be improved by solely refining confidence or not. Note that, our manuscript’s central question is ‘Can we improve self-training for tabular data by making confidence more reliable, without altering the self-training algorithm or model architecture?’.

---

> ### Author Response · Authors · 2023-11-16
> **Official Comment by Authors (2)**
>
> ### Weakness 4
> Limitations are not discussed (*Quality*)
>
> *Answer:* We apologize for the missing limitation of CAST. The manuscript now explicitly states the limitation in the conclusion section that CAST is its inapplicability to domains such as images or text as there are no suitable density estimation methods. The structured nature of tabular data allows for clear density estimation from labeled training datasets as tabular data ensures that each feature occupies a specific, fixed position within the table. But there are no such characteristics in other data types.
>
> ---
>
> ### Question 1
> Can you please elaborate on the 4 above-mentioned weaknesses?
>
> *Answer:* Please see the above answers.
>
>
> ### Question 2
> Regarding experiments on toy datasets, is there any reason why temperature scaling is not shown?
>
> *Answer:* The purpose of the experiments with toy dataset is to demonstrate the ineffectiveness of calibrated confidence in self-training context. We conclude that a single calibration method suffices for the presentation, and we select histogram binning since it is simple and has a lower ECE (Expected Calibration Error) error than temperature scaling.
>
>
> ### Question 3
> In almost all experiments there is a significant difference between the two proposed ways of estimating the density (CAST-D and CAST-L). Can you please discuss about this aspect? Is this issue related to an improper hyperparameter tuning?
>
> *Answer:* The observed difference stems from distinct implementations. For example, CAST-D uses the Aitchison-Aitken kernel for categorical features, while CAST-L uses the likelihood of each distinct value from their empirical distribution. This is not related to improper hyperparameter tuning, as we conducted an exhaustive grid search for optimal alpha selection.
>
>
> ### Question 4
> In Figure 5, can you please explain why the performance decrease with a larger amount of labeled examples, as this seems a counterintuitive result?
>
> *Answer:* The observed performance decrease with a larger amount of labeled examples aligns with findings from several studies [3,4,5]. It illustrates that semi-supervised learning improvements often inversely correlate with labeled sample proportions, which is a commonly observed phenomenon. Therefore, it is not counterintuitive result of our work.
>
>
> [1] Amini, Massih-Reza, and Patrick Gallinari. "Semi-supervised logistic regression." ECAI. Vol. 2. No. 4. 2002.
>
> [2] Berthelot, David, et al. "Mixmatch: A holistic approach to semi-supervised learning." Advances in neural information processing systems 32 (2019).
>
> [3] Rizve, Mamshad Nayeem, et al. "In defense of pseudo-labeling: An uncertainty-aware pseudo-label selection framework for semi-supervised learning." arXiv preprint arXiv:2101.06329 (2021).
>
> [4] Yang, Lihe, et al. "St++: Make self-training work better for semi-supervised semantic segmentation." Proceedings of the IEEE/CVF Conference on Computer Vision and Pattern Recognition. 2022.
>
> [5] Xu, Ran, et al. "Neighborhood-regularized self-training for learning with few labels." arXiv preprint arXiv:2301.03726 (2023).
>
> ---
>
> We hope these comprehensive responses address your concerns and provide a clearer understanding of our research. We sincerely appreciate your guidance and are open to further feedback.

---

> > ### Comment · Reviewer_QLpL · 2023-11-19
> > **Thanks for Answer, Major Concerns are Still Not Addressed**
> >
> > Dear Authors,
> >
> > first of all, thank you for the answers and the clarifications.
> > After going through the rebuttal, I feel there are still unaddressed concerns. Please find below additional questions (also considering the new experiments):
> > - (**Experiment/Stability/Significance**) Why in Table 11 on the sick dataset does CAST completely fail? This makes me wonder about the stability of the results. Could you please estimate the variance of the results and provide the confidence intervals for at least the tables in the main paper?
> > - (**Experiment/Time comparison**) Regarding time, can you please show how does the approach scale in time over dimensions (also on the higher dimensional dataset bioresponse)? Moreover, the comparison should be conducted with the other baselines you have considered in Table 1. I can imagine that temperature scaling has a bigger advantage in terms of computation over the proposed solution. Then, what it the real advantage of the proposed solution in practice?
> > - (**Scope/Significance**) Can you please elaborate more on the weakness 4, as I don’t understand what you mean by “inapplicability to domains such as images or text as there are no suitable density estimation methods”. As far as I know, there are ways to learn the density on both images and text, what about the area of generative models? Also, if the claim about the limited applicability of the approach is true, the results would have little scope and marginal significance.
> > - (**Experiments/Quality**) While I appreciate the discussion about entropy minimisation in semi-supervised learning. I don't see why an experimental comparison is ruled out by that. Are there no semi-supervised approaches for tabular data?

---

> > > ### Author Response · Authors · 2023-11-22
> > >
> > > Thank you for your continued engagement and the additional questions posed regarding our manuscript. We have taken your feedback seriously and have sought to address each point. Below are our responses to your concerns:
> > >
> > > ### 1. (Experiment/Stability/Significance)
> > > - CAST-D fails to improve FPL-based self-training on sick dataset. However, we do not think CAST, regularizing confidence to improve self-training, completely fail. For example, CAST-L achieves improvement unlike CAST-D. The failure might be caused because the combination of Gaussian kernel and Aitchison-Aitken kernel is not appropriate for FPL-based self-training on sick dataset. And we verify CASTs mostly achieve significant improvement on self-training, although there are some cases that some implementations of CAST fail to improve or calibrated confidence based self-training surpass CAST, as shown critical difference diagrams in Figure 4 and the average ranks and standard deviation of ranks of each method in Table 11.
> > >
> > > The table below is the variance, and lower (lb) and upper bound (ub) at 95% confidence level of the results of FPL in Table 1.
> > > | | | | | | 6M |  | mortality | | | | | | | | diabetes | | | | | | | | | ozone | | | | | | | | | cmc | | | | |
> > > | --- | --- | --- | --- | --- | --- | --- | --- | --- | --- | --- | --- | --- | --- | --- | --- | --- | --- | --- | --- | --- | --- | --- | --- | --- | --- | --- | --- | --- | --- | --- | --- | --- | --- | --- | --- | --- | --- |
> > > | | | | xgb | | | ft | | | mlp | | | xgb | | | ft | | | mlp | | | xgb | | | ft | | | mlp | | | xgb | | | ft | | | mlp | |
> > > | | | variance | lb | ub | variance | lb | ub | variance | lb | ub | variance | lb | ub | variance | lb | ub | variance | lb | ub | variance | lb | ub | variance | lb | ub | variance | lb | ub | variance | lb | ub | variance | lb | ub | variance | lb | ub |
> > > | | Baseline | 0.263 | 3.902 | 4.278 | 0.124 | 1.035 | 1.211 | 0.533 | 7.496 | 8.259 | 0.001 | -0.001 | 0.001 | 0.007 | 0.329 | 0.338 | 0.038 | 1.275 | 1.328 | 0.566 | -0.051 | 0.758 | 0.100 | 0.264 | 0.408 | 0.514 | 0.917 | 1.652 | 0.018 | 0.761 | 0.787 | 0.004 | 0.248 | 0.253 | 0.002 | 0.141 | 0.144 |
> > > | | TS | 0.263 | 3.902 | 4.278 | 0.124 | 1.035 | 1.211 | 0.532 | 7.318 | 8.080 | 0.001 | 0.001 | -0.001 | 0.007 | 0.329 | 0.338 | 0.038 | 1.275 | 1.328 | 0.566 | -0.051 | 0.758 | 0.100 | 0.264 | 0.408 | 0.514 | 0.917 | 1.652 | 0.018 | 0.761 | 0.787 | 0.004 | 0.248 | 0.253 | 0.002 | 0.141 | 0.144 |
> > > |  | HB | 0.202 | 3.981 | 4.270 | 0.000 | 0.000  | 0.000 | 0.002 | -0.143 | -0.141 | 0.000 | 0.033 | 0.033 | 0.035 | 0.975 | 1.025 | 0.013 | 0.777 | 0.796 | 0.142 | -0.668 | -0.465 | 0.115 | 2.440 | 2.605 | 0.429 | -0.456 | 0.159 | 0.123 | 0.223 | 0.399 | 0.123 | 0.944 | 1.120 | 0.049 | 0.961 | 1.031 |
> > > FPL | SP | 0.139 | 4.166 | 4.365 | 0.052 | 2.278 | 2.352 | 1.033 | 7.705 | 9.183 | 0.016 | -0.110 | -0.087 | 0.001 | 0.211 | 0.213 | 0.041 | 0.848 | 0.907 | 0.596 | -0.810 | 0.042 | 0.118 | 4.932 | 5.101 | 0.531 | 2.194 | 2.953 | 0.033 | -0.237 | -0.190 | 0.057 | 1.370 | 1.451 | 0.000 | 0.000 | 0.000 |
> > > |  | GP | 0.101 | 1.014 | 1.159 | 0.121 | -1.204 | -1.030 | 0.315 | -0.295 | 0.156 | 0.055 | 0.746 | 0.825 | 0.030 | 0.767 | 0.809 | 0.021 | -0.227 | -0.197 | 1.567 | -0.029 | 2.212 | 0.030 | 0.982 | 1.025 | 1.417 | -2.440 | -0.412 | 0.000 | 0.000 | 0.000 | 0.000 | 0.000 | 0.000 | 0.000 | 0.000 | 0.000 |
> > > |  | CAST-D | 0.104 | 4.017 | 4.166 | 0.259 | 5.377 | 5.747 | 0.560 | 10.142 | 10.943 | 0.048 | 1.570 | 1.638 | 0.040 | 1.365 | 1.422 | 0.033 | 1.701 | 1.749 | 0.406 | 7.040 | 7.621 | 0.421 | 8.559 | 9.161 | 1.132 | 8.245 | 9.865 | 0.044 | 2.293 | 2.357 | 0.027 | 0.696 | 0.735 | 0.068 |1.563 | 1.660 |
> > > |  | CAST-L | 0.190 | 9.462 | 9.733 | 0.198 | 8.809 | 9.093 | 1.167 | 16.146 | 17.816 | 0.031 | 1.320 | 1.364 | 0.020 | 0.652 | 0.681 | 0.062 | 1.923 | 2.011 | 0.471 | 6.251 | 6.925 | 0.177 | 6.603 | 6.855 | 1.469 | 7.005 | 9.107 | 0.032 | 2.341 | 2.386 | 0.094 |1.715 | 1.850 | 0.051 | 1.009 | 1.082 |

---

> > > ### Author Response · Authors · 2023-11-22
> > >
> > > ### 2. (Experiment/Time comparison)
> > > - The time complexity of density estimation is O(N x F) where N is the number of samples and F is the number of features. This can be reduced by some optimized implementations, such as using tree-based structures (like KD-trees or Ball trees) or feature selection methods. Furthermore, we have empirically verified that the computational costs of two CAST implementations are equivalent to the training time of XGBoost shown in Table 10, which is negligible in practice.
> > >
> > > - Although temperature scaling offers computational advantages over the proposed solution, it is meaningless in the context of self-training as shown in Figure 4, the critical difference diagram. Our findings conclude that self-training can be improved by making confidence more reliable, without altering the self-training algorithm or model architecture. And demonstrate that the calibration of confidence is meaningless in the terms of self-training.
> > >
> > > ### 3. (Scope/Significance)
> > > - As you noted, there are several available methods for learning density across other data types, including generative model-based techniques and density peak clustering for images, as well as Uniform Information Density (UID) for text. However, they introduce complexities that deviate from the simplicity inherent in self-training. Concretely, estimate density using generative model require an additional model and UID need a pretrained model [1]. Furthermore, density peak clustering approaches require computational intensity for large datasets (O(N²F)) and potential challenges in automatically identifying cluster centers without manual intervention or visual inspection of the decision graph. Fundamentally, CAST is designed to improve self-training while maintaining simplicity. Therefore, we conclude that there is no suitable density estimation that maintains the advantage of self-training in other data types.
> > >
> > > - And we do not think that our contributions have little scope and marginal significance due to the limited applicability of our approach for other data types. Tabular data have been used in various domains (e.g., medicine, finance, manufacturing, climate science) [2]. Moreover, this is a pioneering work showing that self-training can be improved by solely adjusting confidence to maintain simplicity and versatility without any alterations in self-training algorithm and model architectures. Hence, we believe that CAST opens new avenues for improving self-training methodologies and for practitioners in the field of tabular data.
> > >
> > > ### 4. (Experiments/Quality)
> > > -  Our study aims to answer the following to questions: (1) ‘Can we improve self-training for tabular data by making confidence more reliable, without altering the self-training algorithm or model architecture?’ and (2) ‘Does well-calibrated confidence denote reliable confidence in the self-training context?’. Hence, our research intentionally does not compare CAST with other SSL techniques because our objective is not to challenge the efficacy of these techniques per se, but to demonstrate that self-training can indeed be improved through refined confidence without modifying the self-training algorithms or model architectures. Furthermore, there are some semi-supervised learning (SSL) methods for tabular data such as VIME [3] or contrastive mixup [4], but they are not applicable to GBDTs, which is a dominant architecture for the tabular domain. This is another reason why we do not compare our method with other SSL approaches.
> > >
> > > [1] Venkatraman, et al. "Gpt-who: An information density-based machine-generated text detector.".
> > > [2] Shwartz-Ziv, et al. "Tabular data: Deep learning is not all you need."
> > > [3] Yoon, et al. "Vime: Extending the success of self-and semi-supervised learning to tabular domain."
> > > [4] Darabi, et al. "Contrastive mixup: Self-and semi-supervised learning for tabular domain."
> > >
> > > We hope our responses provided here address your concerns satisfactorily.
> > >
> > > Thank you

---

### Official Review · Reviewer_QRTY · 2023-10-30

**Soundness:** 1 poor
**Presentation:** 3 good
**Contribution:** 2 fair
**Rating:** 3
**Confidence:** 4

**Summary:**

In the paper "CAST: Cluster-Aware Self-Training for Tabular Data" the authors propose an approach to self-training for tabular data. The basic idea of the approach is to take into account how densely populated the dataset is around candidate data points for generating pseudo labels. More specifically, the density is used for regularizing class confidences discounting confidences in less densely populated realms of the space.

**Strengths:**

- The proposed method is relatively simple and thus relatively easy to implement.
- The related work is nicely surveyed.

**Weaknesses:**

- The paper is not self-contained, i.e., there are gaps in the proposed methodology. For example, in Eq. 2 the authors state that prior knowledge is encoded in terms of a vector $\gamma$ and $\gamma$ is assigned the output of some function TD(). However, the function is not clearly described. Only "prior knowledge which is derived from the labeled training data distribution TD"  is mentioned in the text. Probably TD is on purpose quite vague. However, there is not even a single mention of what this could be. It is also confusing that it seems to be the labeled training data distribution, however, this distribution should at maximum be implicitly given by the data sample.
- In Algorithm 1 the $\gamma$ does not even occur. I assume it is somewhere hidden in the $\Phi$ which supposedly does the pseudo-labeling. However, $\Phi$ is nowhere given concretely. Not even in the appendix -- at least I could not find it there. Still, in the text, the authors write that Algorithm 1 is the complete algorithm but only a very basic self-training framework is given there -- nothing special about CAST as a standalone method. Also, the loop is terminating with respect to some unknown termination condition of $\Phi$ which is neither elaborated.
- While the experimental evaluation section covers most of the section, taking different perspectives and viewing angles, the breadth of the study is quite limited. In the main paper, the study comprises 4 real-world datasets with an additional 16 datasets in the appendix for a limited set of methods. Considering that there is no theoretical support for the claims, the underpinning of the claims made in the paper is quite weak.
- Speaking about the empirical evaluation: While relative improvements over a baseline might be the primary goal, with which I agree, it is relatively hard to interpret the significance of the results. In particular, it impedes the application of a statistical test whether the results are significant. From the results, the differences are probably significant but still it is very hard to interpret and I would prefer plain results even though metrics might differ.

**Questions:**

- How is $\gamma$ computed? What are the requirements or desiderata for computing $\gamma$ to yield a sound approach?
- What is the termination criterion in relation to $\Phi$?
- How is the pseudo-labeling $\Phi$ done?

---

> ### Author Response · Authors · 2023-11-16
> **Official Comment by Authors (1)**
>
> Thank you for your thoughtful and detailed feedback on our manuscript. Your comments have been invaluable in improving the clarity and depth of our work.
>
> ---
>
> ### Weakness 1
> The paper is not self-contained, i.e., there are gaps in the proposed methodology. For example, in Eq. 2 the authors state that prior knowledge is encoded in terms of a vector $\gamma$ and $\gamma$ is assigned the output of some function TD(). However, the function is not clearly described. Only "prior knowledge which is derived from the labeled training data distribution TD" is mentioned in the text. Probably TD is on purpose quite vague. However, there is not even a single mention of what this could be. It is also confusing that it seems to be the labeled training data distribution, however, this distribution should at maximum be implicitly given by the data sample.
>
> *Answer:* We apologize for any confusion regarding the density estimation process in our methodology. We have revised the methods section to provide a clearer explanation as follows.
>
> We get the estimated density for unlabeled samples by extracting the prior knowledge using a density estimator $\textit{D}_{t}$ (e.g. multivariate kernel density estimator or empirical likelihood) which is fitted to the labeled training data distribution.
> Here, the prior knowledge $\boldsymbol{\gamma}$ for each class is defined as follows:
>
> $$
> \boldsymbol{\gamma}^{(i)} \leftarrow \textit{D}_{t}(\boldsymbol{x}^{(i)}), \quad \text{where} \quad \boldsymbol{\gamma}^{(i)} = \text{[}{\gamma}_1, \gamma_2, ... , \gamma_N\text{]}
> $$
>
> Then, we normalize $\boldsymbol{\gamma}$ using a min-max scaler because the scale of $\boldsymbol{\gamma}$ varies among implementations, and we need a relative measure to align unlabeled samples.
>
> And we leave the specific choice of density estimator for CAST to implementation as we believe open to extension is a crucial component for the versatility. For example, CAST which uses dataset-specific density estimator might be surpass CAST-D or CAST-L on the specific dataset. Our implementation of CAST-D and CAST-L are shown in Appendix D.
>
> The reason for using the labeled training data distribution in CAST is to enable the regularization of confidence for each class. This is achieved by utilizing class-specific estimated density, which is derived from the labeled data distribution.
>
>
>
> ### Weakness 2
> In Algorithm 1 the $\gamma$ does not even occur. I assume it is somewhere hidden in the $\Phi$ which supposedly does the pseudo-labeling. However, $\Phi$ is nowhere given concretely. Not even in the appendix -- at least I could not find it there. Still, in the text, the authors write that Algorithm 1 is the complete algorithm but only a very basic self-training framework is given there -- nothing special about CAST as a standalone method. Also, the loop is terminating with respect to some unknown termination condition of $\Phi$ which is neither elaborated.
>
> *Answer:* We appreciate your feedback on the lack of clarity regarding $\Phi$ in Algorithm 1. $\Phi$ is the pseudo-labeling algorithm that adopts equation (4), and its specific implementation can vary. CAST is designed to the following question that we raised in the Introduction section: ‘Can we improve self-training for tabular data by making confidence more reliable, without altering the self-training algorithm or model architecture?’. In essence, CAST is an enhanced self-training algorithm which regularizes the confidence of the classifier to be aware of the cluster assumption without altering the self-training algorithm. Therefore, we present the conventional self-training algorithm which adopts regularized confidence for pseudo-labeling as pseudo-code of CAST. And we did not specify a particular $\Phi$ within the pseudo-code, as CAST is designed to be compatible with any self-training algorithm, irrespective of the pseudo-labeler employed. Lastly, the termination condition for $\Phi$ is implementation-specific and hence was not explicitly defined.

---

> ### Author Response · Authors · 2023-11-16
> **Official Comment by Authors (2)**
>
> ### Weakness 3
> While the experimental evaluation section covers most of the section, taking different perspectives and viewing angles, the breadth of the study is quite limited. In the main paper, the study comprises 4 real-world datasets with an additional 16 datasets in the appendix for a limited set of methods. Considering that there is no theoretical support for the claims, the underpinning of the claims made in the paper is quite weak.
>
> *Answer:* We appreciate your insight into the need for a more details are needed to underpin our claims. To address this, we've included the following explanations in Appendix C.
>
> Self-training is a version of the entropy minimization algorithm, which minimizes the likelihood deprived of the entropy of the partition [1]. It constructs hard (one-hot) labels from high-confidence predictions on unlabeled data to implicitly achieve entropy minimization [2]. The entropy minimization techniques assume that the cluster assumption is ensured in the dataset [3], and aim that the classifier learns the low-density separations in the data. However, unreliable pseudo-labels that lie in low-density regions, stemming from erroneous confidence, violate the assumption and consequently disrupt the classifier's ability to learn the separations among classes. On the other hand, CAST forces the pseudo-labels in low-density regions to have lower confidence to avoid the violation of the assumption. Therefore, CAST achieves more reliable pseudo-labels resulting in successful entropy minimization.
>
>
> ### Weakness 4
> Speaking about the empirical evaluation: While relative improvements over a baseline might be the primary goal, with which I agree, it is relatively hard to interpret the significance of the results. In particular, it impedes the application of a statistical test whether the results are significant. From the results, the differences are probably significant but still it is very hard to interpret and I would prefer plain results even though metrics might differ.
>
> *Answer:* We understand the difficulty in interpreting relative improvements and have now included absolute performance in Table 1 and Table 11 in Appendix J. Additionally, we applied statistical tests to show that the performance improvements of CAST are not coincidence, whereas the calibrated confidence based self-trainings are. Concretely, there are some experiments that calibrated confidence based self-training outperform naive confidence based self-training, but statistical test verifies that calibrating the confidence is meaningless in the context of self-training.

---

> ### Author Response · Authors · 2023-11-16
> **Official Comment by Authors (3)**
>
> ### Question 1
> How is $\gamma$ computed? What are the requirements or desiderata for computing $\gamma$ to yield a sound approach?
>
> *Answer:* We get $\gamma$ using a density estimator $\textit{D}_{t}$ which is fitted to the labeled training data distribution. The natural characteristic of the tabular data is each feature occupies a specific, fixed position within the table. This allows us directly to extract prior knowledge from the labeled training dataset unlike other domains (e.g., image or text) where do not have such rules. There are various successful approaches regarding density estimation for tabular data [4,5]. We extract prior knowledge using multivariate kernel density estimator and empirical likelihood.
>
> ### Question 2
> What is the termination criterion in relation to $\Phi$?
>
> *Answer:* The termination criterion depends on the implementation of $\Phi$. For example, if the $\Phi$ is fixed-threshold pseudo-labeling, the termination condition is when no performance gain is achieved. On the other hand, if the $\Phi$ is curriculum pseudo-labeling, it is when there are no unlabeled data left to label.
>
> ### Question 3
> How is the pseudo-labeling $\Phi$ done?
>
> *Answer:* The pseudo-labeler $\Phi$ can be any algorithm suitable for pseudo-labeling, such as fixed-threshold pseudo-labeling or curriculum pseudo-labeling as we designed CAST to maintain versatility. It depends on the implementation of the user. For example, within fixed-threshold pseudo-labeling strategies, pseudo-labels are designated once their confidences meet or exceed a certain threshold. Meanwhile, curriculum pseudo-labeling strategies generate pseudo-labels based on a threshold but operate under the premise that samples with higher confidence are easier for the classifier to handle.
>
>
> [1] Amini, Massih-Reza, and Patrick Gallinari. "Semi-supervised logistic regression." ECAI. Vol. 2. No. 4. 2002.
>
> [2] Berthelot, David, et al. "Mixmatch: A holistic approach to semi-supervised learning." Advances in neural information processing systems 32 (2019).
>
> [3] Grandvalet, Yves, and Yoshua Bengio. "Semi-supervised learning by entropy minimization." Advances in neural information processing systems 17 (2004).
>
> [4] Devroye, Luc. A course in density estimation. Birkhauser Boston Inc., 1987.
>
> [5] Silverman, Bernard W. Density estimation for statistics and data analysis. Routledge, 2018.
>
>
> ---
>
> We hope these revisions provide clarity and address your concerns. We are grateful for your guidance and remain open to any further feedback.

---

### Official Review · Reviewer_DTs7 · 2023-10-31

**Soundness:** 3 good
**Presentation:** 2 fair
**Contribution:** 2 fair
**Rating:** 5
**Confidence:** 3

**Summary:**

This paper aims to use self-training to handle the tabular data learning without altering the self-training algorithm or model architecture.

**Strengths:**

- This paper delves into the confidences of pseudo-labels in self-training from the perspective of cluster assumption, providing a new view for the field of self-training. In addition, the proposed CAST is easy to follow.

**Weaknesses:**

- From the Introduction, I cannot get the significant relationship between tabular data and the proposed self-training method. The motivation and organization of this paper should be further clarified.
- In addition, the difficulties brought by tabular data over the general unstructured data (e.g., images, texts) in machine learning have not been discussed. In detail, they only stated that the GBDT is suitable for tabular data, while they do not explain why other methods are not suitable.
- In my view, CAST is not a tabular data-specific method. Its idea is to adjust the prediction confidence based on the cluster assumption, which is also available for other data types. I think that this discussion should be included and the corresponding experiments are needed.

**Questions:**

Please refer to Weaknesses.

---

> ### Author Response · Authors · 2023-11-16
>
> Thank you for your valuable feedback on our manuscript. We have carefully considered your comments and have revised our manuscript accordingly to address the concerns raised.
>
> ---
>
> ### Weakness 1 and 2
> From the Introduction, I cannot get the significant relationship between tabular data and the proposed self-training method. The motivation and organization of this paper should be further clarified. In addition, the difficulties brought by tabular data over the general unstructured data (e.g., images, texts) in machine learning have not been discussed. In detail, they only stated that the GBDT is suitable for tabular data, while they do not explain why other methods are not suitable.
>
> *Answer:* We acknowledge the need for a clearer exposition of the relationship between tabular data and our proposed self-training method. To address this, we clarify why we study the primary question, ‘Can we improve self-training for tabular data by making confidence more reliable, without altering the self-training algorithm or model architecture?’, in this paper for tabular data, specifically in the second paragraph of the Introduction as follows:
>
> Various studies have proposed solutions to counteract the noise in pseudo-labels induced by erroneous confidence, but they have diminished the simplicity and versatility of self-training. Concretely, they often necessitate modifications to self-training algorithms or alterations in the model architectures (Li & Zhou, 2005; Tanha et al., 2017; Rizve et al., 2021; Seibold et al., 2022). Furthermore, most of them are not applicable to gradient boosting decision trees (GBDT) as they are designed for neural networks. These limitations pose a substantial impediment to practitioners who want to apply reliable self-training on the tabular data where GBDTs have been the dominant architectures (Kaggle, 2021; Borisov et al., 2022; Shwartz-Ziv & Armon, 2022). Therefore, we conclude that any enhanced self-training for the tabular domain must maintain simplicity and versatility. Consequently, we study a natural but ignored question: *Can we improve self-training for tabular data by making confidence more reliable, without altering the self-training algorithm or model architecture?*
>
>
> ### Weakness 3
> In my view, CAST is not a tabular data-specific method. Its idea is to adjust the prediction confidence based on the cluster assumption, which is also available for other data types. I think that this discussion should be included, and the corresponding experiments are needed.
>
> *Answer:* We agree with your observation that the core idea of CAST, which involves adjusting the prediction confidence based on the cluster assumption, could extend to data types beyond tabular data. However, as we now note in our conclusion, the current implementation of CAST is limited by the absence of suitable density estimation methods for non-tabular data types, such as images or text. The structured nature of tabular data allows for clear density estimation from labeled training datasets as tabular data ensures that each feature occupies a specific, fixed position within the table. But there are no such characteristics in other data types. This limitation is now explicitly acknowledged in our manuscript.
>
> ---
>
> We hope these revisions and clarifications address your concerns and enhance the understanding of our research's contributions and limitations. We are grateful for the opportunity to improve our work and welcome any further suggestions you may have.

---

> > ### Comment · Reviewer_DTs7 · 2023-11-23
> >
> > Thank you for your detailed rebuttal. I have gone through your feedbacks and other reviewers' comments. I think I indeed misunderstand your method. I am willing to raise my score to 5 and discuss with other reviewers for further decision.

---

> > > ### Author Response · Authors · 2023-11-23
> > >
> > > Thank you for your willingness to re-evaluate our paper and for considering a score increase. We appreciate your openness and the time you've dedicated to understanding our methodology. If you need any more information during your discussions, please feel free to reach out.
> > >
> > > best regards,
> > >
> > > the authors

---

### Official Review · Reviewer_USSG · 2023-11-01

**Soundness:** 2 fair
**Presentation:** 2 fair
**Contribution:** 2 fair
**Rating:** 5
**Confidence:** 4

**Summary:**

This pager proposes a simple way to generate reliable pseudo-labels by assigning high confidence to pseudo-labels in high-density regions and low confidence to those in low-density regions. The proposed method could be plugged into current self-training algorithms and tabular models, and extensive experiments validate the effectiveness of this method. However, there lacks detailed analysis (empirical or theoretical) or insights on why it works, and the reliable pseudo-labels are not adequately verified in real scenarios.

**Strengths:**

This work proposes a simple but effective way to generate reliable pseudo-labels, which multiples the original pseudo labels with a density score. The method is simple and could be incorporated with various existing algorithms, and extensive experiments validate the effectiveness of it.

**Weaknesses:**

After reading this paper carefully, I have some concerns:
1. There lacks detailed analysis on why this method works. And I wonder whether it has some relationship with label smoothing techniques. And I would recommend the authors to give more in-depth analysis, either empirical or theoretical. Also, the 'cluster assumption' or the reliable pseudo-labels should be checked or verified in real datasets. For example, why they are reliable and can we explain it?
2. I believe there are many many semi-supervised learning methods, but there are only 5 baselines, which I think is not enough and representative for SSL.
3. Some of the datasets are not open-sources, e.g., 6M mortality.

**Questions:**

Please refer to Weaknesses.

---

> ### Author Response · Authors · 2023-11-16
>
> Thank you for the time and effort you have invested in reviewing our manuscript. We greatly appreciate your insightful comments and suggestions, which have provided us with valuable perspectives on our work.
>
> ---
>
> ### Weakness 1
> There lacks detailed analysis on why this method works. And I wonder whether it has some relationship with label smoothing techniques. And I would recommend the authors to give more in-depth analysis, either empirical or theoretical. Also, the 'cluster assumption' or the reliable pseudo-labels should be checked or verified in real datasets. For example, why they are reliable and can we explain it?
>
> *Answer:* We appreciate your insight on the need for a more comprehensive analysis of CAST's efficacy. To address this, we've included the following explanations in Appendix C.
>
> Self-training is a version of the entropy minimization algorithm, which minimizes the likelihood deprived of the entropy of the partition [1]. It constructs hard (one-hot) labels from high-confidence predictions on unlabeled data to implicitly achieve entropy minimization [2]. The entropy minimization techniques assume that the cluster assumption is ensured in the dataset [3], and aim that the classifier learns the low-density separations in the data. However, unreliable pseudo-labels that lie in low-density regions, stemming from erroneous confidence, violate the assumption and consequently disrupt the classifier's ability to learn the separations among classes. On the other hand, CAST forces the pseudo-labels in low-density regions to have lower confidence to avoid the violation of the assumption. Therefore, CAST achieves more reliable pseudo-labels resulting in successful entropy minimization.
>
> And Label smoothing employs soft labels rather than hard labels, such as one-hot encoded vectors, with the aim of preventing the model from becoming overly confident in its predictions. In contrast, CAST specifically aims to reduce the confidence of pseudo-labels in low-density regions, ensuring that unreliable pseudo-labels are not generated. The labels used in CAST remain hard labels.
>
> Lastly, we demonstrate that pseudo-labels in high-density regions are more accurate than those in low-density regions in Figure 1 using 6-month mortality prediction post-acute myocardial infarction (6M mortality) dataset which is sourced from Korea Acute Myocardial Infarction Registry (KAMIR). The Figure 1 shows the heuristic evidence supporting for the cluster assumption.
>
>
> ### Weakness 2
>  I believe there are many many semi-supervised learning methods, but there are only 5 baselines, which I think is not enough and representative for SSL.
>
> *Answer:* We acknowledge the extensive variety of semi-supervised learning (SSL) methods available such as VIME [4]. However, most of the people have used GBDTs for tabular data which are not compatible with most of SSL approaches. Therefore, we focus on self-training which is universal applicable approach and study ‘Can we improve self-training for tabular data by making confidence more reliable, without altering the self-training algorithm or model architecture?’. Hence, our research intentionally does not compare CAST with other SSL techniques because our objective is not to challenge the efficacy of these techniques per se, but to demonstrate that self-training can indeed be improved through refined confidence without modifying the self-training algorithms or model architectures.
>
>
> ### Weakness 3
> Some of the datasets are not open-sources, e.g., 6M mortality.
>
> *Answer:* We understand the concern regarding the use of the non-open source 6M mortality dataset. To balance this, we incorporated 19 open-source datasets from OpenML-CC18, providing a broad and diverse evaluation of CAST. These datasets, coupled with the 6M mortality dataset, offer a robust assessment of CAST’s performance and versatility. The consistent results across these varied datasets confirm CAST's effectiveness and its applicability to a wide range of scenarios.
>
>
> [1] Amini, Massih-Reza, and Patrick Gallinari. "Semi-supervised logistic regression." ECAI. Vol. 2. No. 4. 2002.
>
> [2] Berthelot, David, et al. "Mixmatch: A holistic approach to semi-supervised learning." Advances in neural information processing systems 32 (2019).
>
> [3] Grandvalet, Yves, and Yoshua Bengio. "Semi-supervised learning by entropy minimization." Advances in neural information processing systems 17 (2004).
>
> [4] Yoon, Jinsung, et al. "Vime: Extending the success of self-and semi-supervised learning to tabular domain." Advances in Neural Information Processing Systems 33 (2020): 11033-11043.
>
> ---
>
> We hope that our responses adequately address your concerns and demonstrate the value and robustness of our work. We look forward to any further suggestions or comments you may have.

---

> > ### Comment · Reviewer_USSG · 2023-11-23
> > **Thank you**
> >
> > Thanks for your rebuttal. Some of my concerns are addressed, and I would like to keep my score.

---

> > > ### Author Response · Authors · 2023-11-23
> > >
> > > Thank you for your consideration of our rebuttal. We appreciate the time you've taken to review our responses to your concerns. If there are any further details or clarifications we can provide to assist with your evaluation, please let us know.
> > > We're committed to improving our work and value your feedback.

---

### Author Response · Authors · 2023-11-16
**Global Response for reviewers**

We are profoundly grateful for the time and effort you have dedicated to reviewing our manuscript. We have updated our manuscript to resolve your concerns. Your insightful feedback and constructive critiques have been invaluable in enhancing the quality and clarity of our work. We have carefully considered each point raised and have made thorough revisions to our manuscript to address your concerns. We kindly request that you review our updated manuscript, which now includes the following key changes and clarifications:

1.	*Methodology Clarification*: We have expanded and refined the explanation of CAST, providing a clearer description of how the prior knowledge $\gamma$ is computed.

2.	*Underpinning our claims*: To strengthen the foundation of our claims, we have added detailed explanations in Appendix C, discussing how CAST aligns with the principles of entropy minimization in self-training and effectively addresses the reliability of pseudo-labels.

3.	*Empirical Evaluation and Statistical Significance*: We have included absolute performance metrics in our manuscript and detailed explanations of the statistical tests which were conducted to assess the performance of CAST.

4.	*Computational Cost of CAST*: Recognizing the importance of practicality in the application of our method, we have included a detailed analysis of CAST’s computational cost in Appendix H. This analysis demonstrates that CAST’s computational demands are comparable to those of the training time of XGBoost and significantly lower than neural network-based methods, underlining its efficiency and feasibility for practical use.

5.	*Limitation*: We have explicitly included the limitation of CAST in the conclusion, its current applicability to tabular data and the challenges in extending it to other data types.

We believe these revisions comprehensively address the concerns raised in your reviews and significantly enhance the manuscript. We are grateful for your valuable input and would greatly appreciate your review of the updated manuscript to consider these enhancements.

Thank you once again for your invaluable contributions to our research.

Best regards,

The authors

---

### Meta-Review · Area_Chair_jguq · 2023-12-09

**Metareview:**

In this paper, the reviewers see the merits of the proposed algorithm for self-training for tabular data and how it compares concerning baselines. Still, the paper has several elements of improvement to be presented at a major conference: Computational complexity from the density estimation and how it will perform in higher dimensional spaces and limited theory.

I can also add that the literature on self-supervised learning is vast, the authors only mentioned a couple of papers, and there are no comparisons with semisupervised learning algorithms. Besides Olivier Chapelle, Jerry Zhu (https://scholar.google.com/citations?user=hqTu-QcAAAAJ&hl=en) and Mikhail Belkin (https://scholar.google.com/citations?user=Iwd9DdkAAAAJ&hl=en) have significant papers on the topic that are not cited or compared to in this paper. The comparisons are with supervised learning algorithms that do not use unlabeled data.

**Justification For Why Not Higher Score:**

The paper does not cite significant works in semi-supervised learning or compares with them.

**Justification For Why Not Lower Score:**

there is nothing higher

---

### Decision · Program_Chairs · 2024-01-16

Reject